

# An Improved Representation of Aerosol Mixing State for Air-Quality-Weather Interactions

Robin Stevens[1,2], Andrei Ryjkov[1], Mahtab Majdzadeh[3], and Ashu Dastoor[1]

[1]Air Quality Research Division, Environment and Climate Change Canada, 2121 Trans-Canada Highway, Dorval, Québec, Canada
[2]Université de Montreal, Montréal, Quebec, Canada
[3]Air Quality Research Division, Environment and Climate Change Canada, 4905 Dufferin Street, Toronto, Ontario, Canada

**Correspondence:** Ashu Dastoor (Ashu.Dastoor@ec.gc.ca)

**Abstract.** We implement a detailed representation of aerosol mixing-state into the GEM-MACH air quality and weather forecast model. Our mixing-state representation includes three categories: one for more-hygroscopic aerosol, one for less-hygroscopic aerosol with a high black carbon (BC) mass fraction, and one for less-hygroscopic aerosol with a low BC mass fraction. This is the first model with a mixing-state representation of this type simulating a continent-scale domain. The more-

detailed representation allows us to better resolve two different aspects of aerosol mixing state: differences in hygroscopicity due to aerosol composition, and the amount of absorption enhancement of BC due to non-absorbing coatings. Notably, this three-category representation allows us to account for BC thickly coated with primary organic matter, which enhances the absorption of the BC but has a low hygroscopicity.

We compare the results of the three-category representation (1L2B) with a simulation that uses two categories, split by hy-

groscopicity (HYGRO), and a simulation using the original size-resolved internally mixed assumption (SRIM). We find that the more-detailed representation of the aerosol hygroscopicity in both 1L2B and HYGRO decreases wet deposition, which increases aerosol concentrations, particularly of less-hygroscopic species. The concentration of PM$_{2.5}$ increases by 23% on average. We show that these increased aerosol concentrations increase cloud droplet number concentrations and cloud reflectivity in the model, decreasing surface temperatures.

Using two categories based on hygroscopicity yields only a modest benefit in resolving the coating thickness on black carbon, however. The 1L2B representation resolves BC with thinner coatings than the HYGRO simulation, resulting in absorption aerosol optical depths that are 3% less on average. We did not find strong subsequent effects of this decreased absorption on meteorology.

## 1 Introduction

Aerosol chemical mixing state refers to the distribution of chemical species across a population of aerosol particles. An aerosol population is said to be fully externally mixed if each aerosol particle consists of a single chemical species. If all chemical species are distributed evenly amongst all aerosol particles, then the aerosol population is said to be fully internally mixed. Aerosol populations in the real atmosphere are never fully externally mixed nor fully internally mixed, but instead exist some-



where between these two extremes. In general, particles emitted from different sources are initially externally mixed with
respect to each other, and become more internally mixed with time through condensation, coagulation, and chemical reactions.

Internal mixing of hydrophilic and hydrophobic species can allow the hydrophobic species to act as cloud condensation
nuclei (CCN) (e.g. McFiggans et al., 2006; Anttila, 2010; Kim et al., 2018; Dalirian et al., 2018). An increase in CCN con-
centrations will generally render clouds more reflective, and can also increase cloud lifetime. Internal mixing of hydrophilic
and hydrophobic species also allows the hydrophobic species to be more efficiently removed from the atmosphere through wet
deposition. Additionally, weakly absorbing species can form a coating on black carbon (BC), which strongly absorbs solar
radiation. The weakly absorbing species then act as a lens, enhancing the absorption of solar radiation by the BC, compared to
the case where the BC is uncoated (e.g. Lesins et al., 2002; Liu and Mishchenko, 2018; Schnaiter et al., 2005; Cui et al., 2016).
Estimates of the factor by which the absorption of BC increases due to coatings (the absorption enhancement) vary from 1 (no
enhancement) to 4, with the majority of the studies reporting values between 1 and 2.5 (Adachi et al., 2010; Zhang et al., 2008;
Khalizov et al., 2009; Cappa et al., 2012; Lack et al., 2012; Liu et al., 2015; Peng et al., 2016; Schnaiter et al., 2005; Wang
et al., 2014a; Xu et al., 2018; Zanatta et al., 2018; Zhang et al., 2018b). Differences in experimental methods and regional
and seasonal variations in BC coating thickness both likely contribute to this diversity. This absorption enhancement leads to
a local heating of the atmosphere and a cooling of the surface, potentially increasing stability and affecting cloud cover and
precipitation (Bond et al., 2013; Boucher et al., 2013). We refer the reader to two recent reviews (Stevens and Dastoor, 2019;
Riemer et al., 2019) for more details about aerosol mixing state.

Many previous representations of aerosol mixing-state have been implemented in models to predict CCN concentrations
and aerosol optical properties. These include representing each particle individually (PartMC-MOSAIC, Riemer et al., 2009;
Zaveri et al., 2010); multiple mixing-state categories separated by BC mass fraction, including MADRID-BC (Oshima et al.,
2009b, a), ATRAS (Matsui et al., 2014; Matsui, 2017), MADE-soot (Riemer et al., 2003; Vogel et al., 2009), MADE-in (Aquila
et al., 2011) and MADE-3 (Kaiser et al., 2019, 2014); two categories for at least BC and organic carbon based on hygroscop-
icity, implemented in GEOS-Chem (Bey et al., 2001; Wang et al., 2018, 2014b), the GLObal Model of Aerosol Processes
(GLOMAP) in both its bin (Manktelow et al., 2010; Spracklen et al., 2005, 2011) and modal (Mann et al., 2010; Bellouin
et al., 2013) configurations, GMXe (Pringle et al., 2010), the M3+ module (Wilson et al., 2001), M7 (Vignati et al., 2010;
Stier et al., 2005; Vignati et al., 2004; Zhang et al., 2012), MAM4 (Liu et al., 2016), MAM7 (Liu et al., 2012), the Model
for Ozone and Related chemical Tracers (MOZART, Emmons et al., 2010) and the Sectional Aerosol module for Large Scale
Applications (SALSA, Bergman et al., 2012; Andersson et al., 2015; Kokkola et al., 2008; Tonttila et al., 2017; Kokkola et al.,
2018); and representing all aerosol within the same size bin or mode as internally-mixed, including the Canadian Aerosol
Module (CanAM, Gong et al., 2006; Moran et al., 2012; Gong et al., 2003, 2015), CHIMERE (Menut et al., 2013), the Com-
munity Multiscale Air Quality (CMAQ, Binkowski and Roselle, 2003; Appel et al., 2013; Elleman and Covert, 2009; USEPA,
2017) model, the Modal Aerosol Dynamics module for Europe (MADE, Lauer et al., 2005) and the Modal Aerosol Module
with three lognormal modes (MAM3, Liu et al., 2012). We refer the reader to Stevens and Dastoor (2019) for more detail



on previous model representations of aerosol mixing state, including mixing-state representations that did not specifically tar-
get resolving CCN concentrations and optical properties, such as detailed categorizations based on chemical composition and
source-oriented approaches. Previous studies using the model approaches listed above have found that if all aerosol in the same
size bin or mode is assumed to be internally-mixed, CCN concentrations will frequently be overestimated by 10-20 % and
absorption coefficients of BC will be overestimated by 20-40 % (Stevens and Dastoor, 2019, and containing references).

However, it still remains unclear how best to efficiently represent aerosol mixing state in atmospheric models. In this study,
we implement a detailed representation of aerosol mixing-state into the Global Environmental Multiscale - Modelling Air
quality and CHemistry (GEM-MACH) (Moran et al., 2010) air quality model with online air-quality-weather interactions. We
refer to this new configuration of GEM-MACH as GM-MixingState. Our approach was inspired by the results of Ching et al.
(2016): We independently account for both changes in hygroscopicity and BC mass fraction, as aerosol hygroscopic proper-
ties and optical properties do not necessarily co-vary. The existing air-quality-weather interactions in GEM-MACH include
aerosol-radiation interactions and changes in cloud droplet activation based on CCN concentrations (Gong et al., 2015; Ma-
jdzadeh et al., 2022). We perform a case study focused on biomass-burning over North America to evaluate GM-MixingState.
We investigate the interactions between the representation of aerosol mixing state and air-quality-weather interactions.

The paper is structured as follows: In Sect. 2, we describe the GEM-MACH model and the GM-MixingState configuration,
as well as the experiments performed. In Sect. 3, we present our results and analysis. In Sect. 4, we summarize our study and
present our conclusions.

## 2 Model description and methods

GEM-MACH is an online chemical transport model embedded within the Environment and Climate Change Canada (ECCC)
Numerical Weather Prediction (NWP) model GEM (Côté et al., 1998b, a; Charron et al., 2012). GEM-MACH has been in use
as the ECCC operational air quality prediction model since 2009 (Moran et al., 2010). The representations of many atmospheric
processes in GEM-MACH are the same as in the ECCC AURAMS (A Unified Regional Air-quality Modelling System) offline
chemical transport model (Gong et al., 2006), including gas-phase, aqueous-phase, and heterogeneous chemistry (inorganic
gas-particle partitioning); secondary organic aerosol (SOA) formation; aerosol microphysics (nucleation, condensation, coagu-
lation, and activation); sedimentation of particles; and dry deposition and wet removal (in-cloud and below-cloud scavenging)
of gases and particles. Eight dry aerosol chemical species are included in GEM-MACH: sulphate, nitrate, ammonium, sea-salt,
dust and crustal material, SOA, primary organic aerosol (POA), and BC. We note that we will refer to dust and crustal material
collectively as "dust" in the rest of this paper.

By default, GEM-MACH uses a size-resolved internally mixed representation of the aerosol population: the aerosol popula-
tion within each size bin is internally mixed, but the population of aerosol in each size bin is externally mixed with respect to



each other size bin. The operational version of GEM-MACH uses two size bins (0-2.5 $\mu$m and 2.5-10 $\mu$m, Moran et al. (2010)), but for this study we use twelve size bins spanning 10 nm to 10 $\mu$m. The 12-bin configuration has been shown to yield results that more closely resemble observations (Akingunola et al., 2018).


For this study, we implemented a more detailed representation of the aerosol mixing state into GEM-MACH. Within each size bin, we separate the aerosol into up to three mixing-state categories based on hygroscopicity and BC mass fraction: 1. high hygroscopicity (hi-$\kappa$); 2. low hygroscopicity, high BC mass fraction (lo-$\kappa$_hi-BC); and 3. low hygroscopicity, low BC mass fraction (lo-$\kappa$_lo-BC). This configuration is similar to the MADE-soot, MADE-in and MADE-3 aerosol modules, which

include three categories: generally hydrophilic BC-free particles, hydrophilic BC-containing particles and hydrophobic BC-containing particles. We differ in that we our BC-free category is also hydrophobic, and we have a single category for all hydrophilic particles. This allows us to resolve BC coated with organic material (weakly hygroscopic, but thickly-coated) from BC coated with hydrophilic material, which is both hydrophilic and thickly-coated. Following the recommendations in Ching et al. (2016), we use a threshold value of the hygroscopicity parameter ($\kappa$; Petters and Kreidenweis (2007)) of 0.1 between hi-$\kappa$

and lo-$\kappa$ mixing-state categories, and a threshold BC mass fraction of 0.3 between lo-BC and hi-BC mixing-state categories. We will discuss these mixing-state categories further in Sect. 2.2.

Coagulation of two particles within the same mixing-state category is assumed to result in a particle of the same mixing-state category, as both BC mass fraction and volume-weighted hygroscopicity would be within the range spanned by the two

original particles. For coagulation of particles from two different mixing-state categories, we calculate the hygroscopicity and BC mass fraction of the new particle, and add the mass to the mixing-state category that matches the new particle's properties. Additionally, after all other aerosol processes, we calculate the hygroscopicity and BC mass fraction for each size bin and mixing-state category. If either the hygroscopicity or the BC mass fraction is outside of the bounds of the current mixing-state category, the mass is moved to the mixing-state category that matches the hygroscopicity and BC mass fraction of the aerosol

mass.

To calculate the hygroscopicity of aerosol in the model, we assume that sulphate, nitrate, ammonium, sea-salt, dust, SOA, POA, and BC have $\kappa$ values of 0.65, 0.65, 0.65, 1.1, 0.03, 0.1, 0.001, and 0, respectively (Ching et al., 2016; Zieger et al., 2017; Koehler et al., 2009). Following the volume Zdanovskii-Stokes-Robinson (ZSR) mixing rule (Petters and Kreidenweis,

2007), we assume that the hygroscopicity of a particle is the volume-weighted average of the component species. We therefore do not account explicitly for coating of insoluble components by soluble components, nor do we consider how particle size or shape may affect the mass fraction of coating material necessary for a particle to be rendered "hydrophilic". However, other studies have shown that neither CCN concentrations (Liu et al., 2016; Lee et al., 2013) nor aerosol effective radiative forcing, either through aerosol-cloud interactions or through aerosol-radiation interactions (Regayre et al., 2018), are sensitive to the

threshold amount of soluble material needed to render a particle hydrophilic. However, global burdens of BC and POA, especially in remote regions, have been shown to be sensitive to this parameter (Liu et al., 2012, 2016). This volume-weighted




hygroscopicity is only used to determine the proper mixing-state category for aerosol. It is not used to determine cloud droplet
activation.

Instead, cloud droplet activation is calculated using the parameterization for sectional models described by Abdul-Razzak
(2002). Particle hygroscopicity is calculated separately for each mixing-state category based on molecular weights and ion
dissociation, as per eq. 7 from Abdul-Razzak (2002). Properties of SOA are assumed to be those of adipic acid; BC, POA and
dust are assumed to be insoluble. We assume that aerosol in the lo-$\kappa$ mixing-state categories does not participate in aqueous
chemistry, and is not removed by cloud-to-rain conversion and subsequent wet deposition. It is still removed from the atmo-
sphere by below-cloud impaction by rain, as this process is not expected to depend strongly on aerosol composition.

Aerosol-radiation interactions are calculated as described in Majdzadeh et al. (2022). For the radiation calculations, sea-
spray and dust are always assumed to exist as pure particles, externally mixed from the other components. For each size bin
within each mixing-state category, the remaining components are assumed to form a spherical shell around a spherical BC
core. The absorption enhancement of the BC cores is calculated following Bond et al. (2006). However, we assume that there
is no absorption enhancment for BC cores that comprise more than 40% of the particle by mass, in agreement with more
recent observations of thinly-coated BC particles (Liu et al., 2017; Peng et al., 2016). Therefore, there is never any absorption
enhancement for particles in the hi-BC mixing-state category.

We choose our domain and time period to be consistent with the 2016 North Americain domain used in the fourth phase of
the Air Quality Model Evaluation International Initiative (AQMEII4; Galmarini et al., 2021). To reduce computational expenses
while still providing sufficient data for analysis, we perform simulations only from June 15$^{th}$ to July 31$^{st}$, and we restrict our
analysis to output from the month of July to provide sufficient time for spin-up. We use a 10 km horizontal resolution and 84
hybrid vertical levels up to 0.1 hPa, consistent with the ECCC contribution to AQMEII4 multi-model experiment. Chemical
boundary conditions are sourced from a climatology from the global chemical transport model MOZART-4 (Model for Ozone
and Related chemical Tracers, version 4; Emmons et al., 2010).

## 2.1 Emissions

The emissions inventories used in study are the same as those described in Majdzadeh et al. (2022), and very similar to the
protocol for contributions to AQMEII4 (Galmarini et al., 2021). Anthropogenic emissions from Canada and the United States
were sourced from the Canadian Air Pollutant Emissions Inventory and the US Environmental Protection Agency (EPA) 2011
Air Emissions Modelling Platform, respectively. We use forest fire emissions from the Canadian Forest Fire Emissions Pro-
duction System (CFFEPS; Chen et al., 2019). The Sparse Matrix Operator Kernel Emissions (SMOKE) emissions processing
system (https://www.cmascenter.org/smoke; Bieser et al., 2011; Hogrefe et al., 2003; Houyoux et al., 2000) is used to speciate
emissions prior to input within GEM-MACH (Zhang et al., 2018a). Bulk aerosol mass emissions are associated with one of the
91 composite particulate matter speciation profiles compiled from the EPA's SPECIATE4.5 database (https://www.epa.gov/air-



emissions-modeling/speciate-2; Reff et al., 2009). Each composite particulate matter speciation profile gives relative fractions of sulphate, nitrate, ammonium, BC, and POA. Sea salt and SOA are assumed to make no contribution, and dust is defined to be the residual after the other components are accounted for. As an example, particulate emissions from wildfires are speciated as 78.5% POA, 9.7% dust, 9.5% BC, 1.3% sulphate, 0.9% ammonium, and 0.1% nitrate. Sea-spray emissions are parameterized
according to Gong (2003).

We differ from these previous studies in that we allocate aerosol emissions across the different mixing-state categories, as follows: Major stationary point-source emissions, such as emissions from smelters or fossil-fuel power plants, are assumed to be size-resolved internally mixed with other particulate mass from the same point source. Area emissions, including sea-spray,
dust, and disperse anthropogenic emissions, including traffic emissions, are assumed to be as close to fully externally mixed as possible within the limits of the mixing-state representation used. If all three mixing-state categories were used, sulphate, ammonium, nitrate, and sea-spray would be emitted in the hi-$\kappa$ category; dust and POA would be emitted into the lo-$\kappa$_lo-BC category; and BC would be emitted into the lo-$\kappa$_hi-BC category. There are no primary emissions of SOA. We note that these emissions are not truly fully externally mixed in reality, and that this assumption will provide the maximum sensitivity to
the mixing-state configuration used. However, observations have shown that particles emitted from traffic sources are either primarily BC or primarily organic, rather than being fully internally mixed at emission (Willis et al., 2016). Wildfire emissions are treated separately from other area emissions in GEM-MACH, and we assume that wildfire emissions are emitted into the lo-$\kappa$_lo-BC category category, as BC-containing particles within wildfire emissions have been observed to frequently be thickly coated with low-hygroscopicity organic material (Perring et al., 2017; Kondo et al., 2011).

**2.2 Sensitivity studies**

An important question remains regarding the minimum level of complexity required to well represent aerosol-weather feedbacks in air quality models. We therefore perform several simulations with diverse representations of the aerosol mixing state. We consider a configuration with two mixing-state categories, split based on particle hygroscopicity (denoted as representation HYGRO; categories hi-$\kappa$ and lo-$\kappa$). We also consider a mixing-state representation with three mixing-state categories:
we use one mixing-state category for all high-hygroscopicity particles and two mixing-state categories for low-hygroscopicity particles, split based on BC mass fraction (high-$\kappa$, low-$\kappa$_hi-BC, and low-$\kappa$_lo-BC). We refer to this representation as 1L2B (one hydrophilic, two hydrophobic). We would not expect any improvement over HYGRO in the representation of the radiative properties of hydrophilic particles, but we would expect that 1L2B would better represent the radiative properties of low-hygroscopicity BC-containing particles. In particular, this representation should better distinguish BC thickly coated with
POA from BC that is bare or only thinly coated with POA.

In addition to performing simulations with different representations of the aerosol mixing state, we also perform simulations with aerosol effects on meteorology (feedbacks) either permitted or disabled. When feedbacks are disabled, cloud droplet nucleation is independent of aerosol concentrations and aerosol interactions with radiation have no effect on atmospheric tem-





peratures or any other meteorological variables. Cloud droplet nucleation is instead determined following Cohard and Pinty
(2010) as a function of updraft velocity, temperature, and pressure assuming a pre-specified CCN concentration that does not
vary with space or time. The meteorology in these simulations is independent of the aerosol and gas-phase concentrations.
This allows us to directly attribute any differences in results solely to differences in aerosol processes caused by the differences
in the representation of mixing state. We designate the simulations where aerosol effects on meteorology are permitted with
the suffix "_feedbacks", as these simulations include feedbacks of changes in aerosol concentrations and properties on the
meteorology.

## 3    Results and analysis

### 3.1    Non-Feedbacks Simulations

We present a summary of the domain-averaged, temporally averaged results from all simulations in Table 1. We will start
by discussing differences between simulations with aerosol effects on weather disabled, in order to simplify the analysis. We
remind the reader that because the meteorology is identical in these simulations, any differences in results can be attributed
solely to differences in aerosol processes caused by the differences in the representation of mixing state.

### 3.1.1    Aerosol Concentrations

We show the mean concentrations of particulate matter with a diameter smaller than 2.5 microns ($PM_{2.5}$) in Fig. 1, along with
the absolute and relative differences in $PM_{2.5}$ concentrations between the HYGRO and SRIM simulations, and we show a sim-
ilar figure for $PM_{10}$ concentrations as Fig. S1. We note that $PM_{2.5}$ and $PM_{10}$ concentrations are nearly identical in the HYGRO
and 1L2B simulations. We find that spatially and temporally averaged surface $PM_{2.5}$ concentrations and $PM_{10}$ concentrations
increase by 23% and 41%, respectively, from the SRIM simulation to either the HYGRO or 1L2B simulations. These differ-
ences are due mostly to increases in less-hygroscopic species, with concentrations of BC, POA, SOA, and dust being increased
in the HYGRO and 1L2B simulations by 16%, 16%, 21%, and 93%. The concentrations of more-hygroscopic species ($NH_4$,
$NO_3$, $SO_4$, and sea-spray aerosol) were increased by 3% or less.

These changes in aerosol concentrations are due primarily to changes in aerosol wet deposition. In the HYGRO and 1L2B
simulations, all aerosol in the low-$\kappa$ categories are excluded from wet deposition processes. However, direct comparison of wet
deposition fluxes between simulations is complicated because of the greater aerosol mass concentrations in the HYGRO and
1L2B simulations than in the SRIM simulation. Even though the wet deposition process is less efficient for the same air parcel
under the same conditions, local wet deposition fluxes can be greater in the HYGRO and 1L2B simulations due to the greater
mass concentrations of aerosol in these simulations. For example, a reduced wet deposition flux close to an emissions source



**Table 1.** Temporally and spatially averaged results for each simulation.

| Simulation | SRIM | HYGRO | 1L2B | SRIM_feedbacks | HYGRO_feedbacks | 1L2B_feedbacks |
|---|---|---|---|---|---|---|
| PM$_{2.5}$ [$\mu$g kg$^{-1}$] | 4.64 | 5.71 | 5.72 | 4.67 | 5.70 | 5.70 |
| PM$_{10}$ [$\mu$g kg$^{-1}$] | 5.67 | 7.98 | 7.99 | 5.69 | 7.97 | 7.98 |
| AQHI$_{2.5}$ | 1.36 | 1.40 | 1.40 | 1.34 | 1.38 | 1.38 |
| AQHI$_{10}$ | 1.27 | 1.33 | 1.33 | 1.27 | 1.33 | 1.33 |
| NH$_4$ [$\mu$g kg$^{-1}$] | 0.1572 | 0.1624 | 0.1627 | 0.1598 | 0.1653 | 0.1657 |
| NO$_3$ [$\mu$g kg$^{-1}$] | 0.0251 | 0.0255 | 0.0257 | 0.0239 | 0.0244 | 0.0246 |
| SO$_4$ [$\mu$g kg$^{-1}$] | 0.537 | 0.551 | 0.552 | 0.554 | 0.570 | 0.572 |
| SOA [$\mu$g kg$^{-1}$] | 1.75 | 2.11 | 2.12 | 1.76 | 2.11 | 2.12 |
| POA [$\mu$g kg$^{-1}$] | 0.875 | 1.013 | 1.014 | 0.870 | 0.993 | 0.994 |
| Sea salt [$\mu$g kg$^{-1}$] | 6.89 | 6.93 | 6.93 | 6.89 | 6.99 | 6.99 |
| Dust [$\mu$g kg$^{-1}$] | 1.72 | 3.31 | 3.31 | 1.72 | 3.30 | 3.30 |
| BC [$\mu$g kg$^{-1}$] | 0.150 | 0.175 | 0.175 | 0.150 | 0.173 | 0.172 |
| BC mass fraction* | 2.12% | 4.29% | 13.63% | 2.09% | 4.21% | 13.39% |
| AOD** | 0.0720 | 0.0962 | 0.0959 | 0.0732 | 0.0966 | 0.0964 |
| AAOD** | 0.0074 | 0.0103 | 0.0100 | 0.0076 | 0.0104 | 0.0101 |
| precipitation [mm day$^{-1}$] | 0.001607 | 0.001607 | 0.001607 | 0.001621 | 0.001611 | 0.001612 |

*Average BC mass fraction within BC-containing particles, see Sect. 3.1.4

**Averaged over 1300-2100 UTC daily.

can yield an increased wet deposition flux further downwind, as more aerosol mass will be transported further downwind. We attempt to isolate for these effects by dividing the daily wet deposition flux by the daily mean surface aerosol concentrations, to approximate the wet deposition efficiency. This approach is limited in that cloud uptake of gases also contributes to the wet deposition fluxes, and cloud uptake of aerosol and subsequent wet deposition are not necessarily co-located in space and
time with surface aerosol concentrations. However, we expect that the relationships between surface concentrations and wet deposition fluxes are similar enough across simulations for the comparison between simulations to be informative.

We show the temporal means of the wet deposition fluxes normalized by the surface aerosol concentrations in Fig. 2, along with the absolute and relative differences between the HYGRO and SRIM simulations. Both aerosol concentrations and wet
deposition fluxes are nearly identical in the HYGRO and 1L2B simulations. We note that the normalized wet deposition shows some similar patterns to surface precipitation, as shown in Fig. 10, and the greatest values of normalized wet deposition are in the northern part of the domain where PM$_{2.5}$ concentrations are low (Fig. 1). In the HYGRO simulation, normalized wet deposition fluxes decrease over most of the domain, except for some regions in the south of the model domain. Deeper clouds would be expected in this part of the domain, which may decouple wet deposition fluxes from surface aerosol concentrations.
In the far north of the domain, there are large decreases in normalized wet deposition fluxes, in some cases approaching 100%.



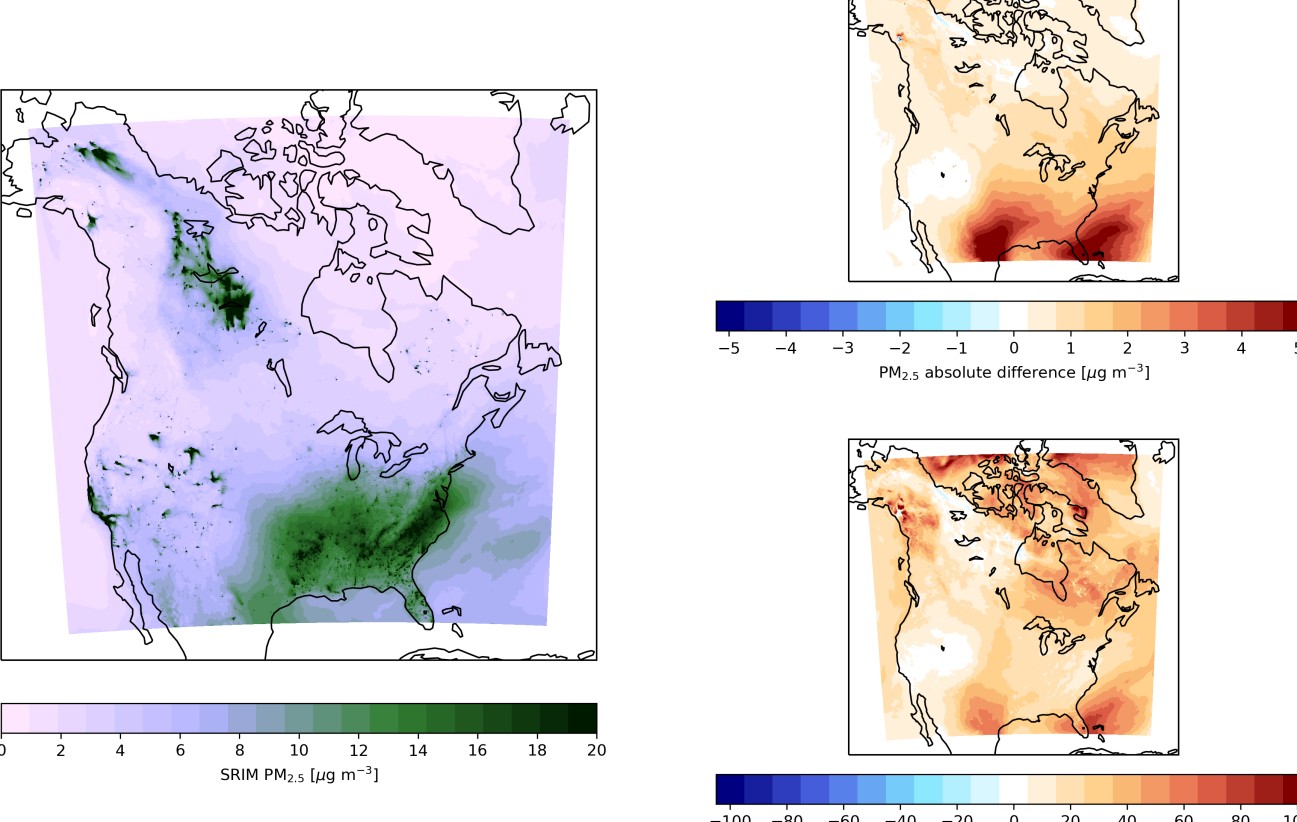

**Figure 1.** left: mean PM$_{2.5}$ concentrations from the SRIM simulation; top right: mean difference in PM$_{2.5}$ concentrations between the HYGRO and SRIM simulations; bottom right: relative difference in mean PM$_{2.5}$ concentrations between the HYGRO and SRIM simulations. Note that PM$_{2.5}$ concentrations are nearly identical in the HYGRO and 1L2B simulations.

These overlap with regions of low aerosol concentrations and large relative increases in surface aerosol concentrations, as shown in Fig. 1. However, other regions with greater mean surface aerosol concentrations and smaller differences in surface aerosol concentrations between the SRIM and HYGRO simulations also show large relative differences in normalized wet deposition fluxes. Normalized wet deposition fluxes over most of Canada are reduced by 20-80%. A large part of this region is influenced by the forest fires that took place in Alaska and northern Canada during this period. As these forest fire emissions are composed primarily of POA, they are particularly sensitive to the changes in the representation of the aerosol mixing state.

We can further control for differences in location and timing between wet deposition and surface concentrations by temporally and spatially averaging both the wet deposition fluxes and the surface concentrations before we divide the former by the



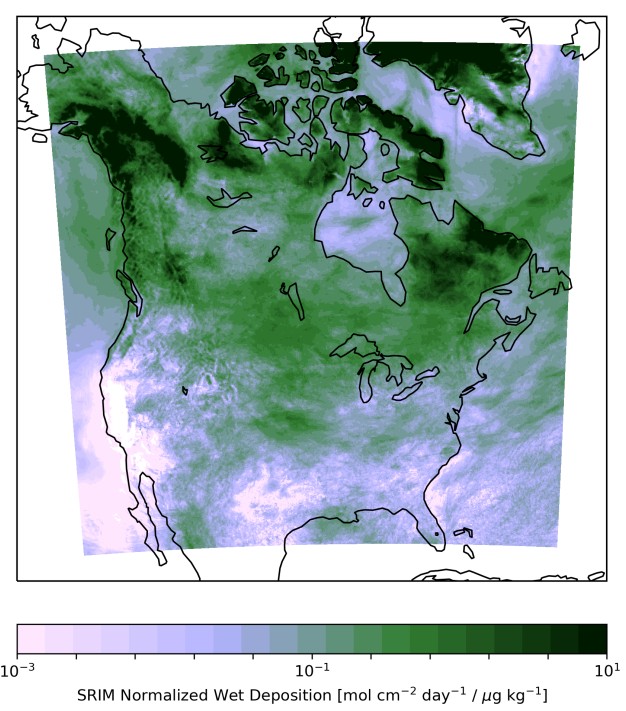

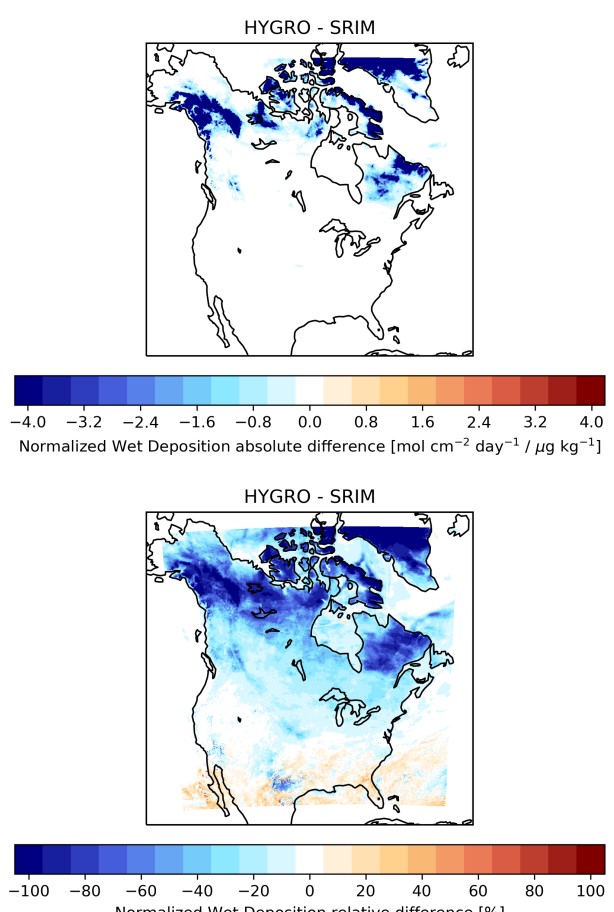

**Figure 2.** left: temporal means of wet deposition fluxes normalized by surface total aerosol concentrations from the SRIM simulation; top right: mean difference in normalized wet deposition fluxes between the HYGRO and SRIM simulations; bottom right: relative difference in mean normalized wet deposition fluxes between the HYGRO and SRIM simulations. Note that wet deposition fluxes and total aerosol concentrations are nearly identical in the HYGRO and 1L2B simulations.

latter. We therefore show the spatially and temporally averaged wet deposition fluxes normalized by the spatially and temporally averaged surface concentrations of each species in Table 2. After normalizing by the surface concentrations of aerosol, wet deposition rates of BC, POA, SOA, and dust were reduced in the HYGRO and 1L2B simulations by 27%, 40%, 12%, and 10%. The normalized wet deposition rates of more-hygroscopic species were reduced by less than 5%.

In the HYGRO and 1L2B simulations, all aerosol in the low-$\kappa$ categories are excluded from wet deposition processes. Since the low-$\kappa$ category is defined as having a $\kappa$ less than 0.1, this excludes large aerosol with low hygroscopicities from participating in wet deposition, even if their large size would allow them activate as droplets despite their low hygroscopicity.



**Table 2.** Temporally and spatially averaged wet deposition fluxes normalized by temporally and spatially averaged surface concentrations for each simulation. All units are (mol cm$^{-2}$ day$^{-1}$) / ($\mu$g kg$^{-1}$).

| Simulation | SRIM | HYGRO | 1L2B | SRIM_feedbacks | HYGRO_feedbacks | 1L2B_feedbacks |
|---|---|---|---|---|---|---|
| NH$_4$ | 0.765 | 0.744 | 0.742 | 0.750 | 0.726 | 0.724 |
| NO$_3$ | 1.547 | 1.533 | 1.525 | 1.618 | 1.588 | 1.577 |
| SO$_4$ | 0.0413 | 0.0394 | 0.0394 | 0.0381 | 0.0362 | 0.0361 |
| SOA | 0.0314 | 0.0276 | 0.0275 | 0.0312 | 0.0277 | 0.0277 |
| POA | 0.0214 | 0.0129 | 0.0129 | 0.0211 | 0.0134 | 0.0133 |
| Sea salt | 0.0727 | 0.0724 | 0.0724 | 0.0735 | 0.0723 | 0.0721 |
| Dust | 0.0526 | 0.0473 | 0.0472 | 0.0525 | 0.0472 | 0.0471 |
| BC | 0.267 | 0.195 | 0.196 | 0.265 | 0.199 | 0.201 |

In particular, this may cause the wet deposition of dust particles to be underestimated in the HYGRO and 1L2B simulations, while it is likely overestimated in the SRIM simulation. A more detailed treatment of cloud uptake of aerosol is beyond the scope of this study, but will be revisited in a future version of GEM-MACH.

### 3.1.2 Comparison with Observations

We compare the results of our non-feedbacks simulations against data from the Interagency Monitoring of Protected Visual Environments (IMPROVE; http://vista.cira.colostate.edu/Improve/, last access 3 March 2022), the US EPA Chemical Speciation Network (CSN), and hourly measurements of PM$_{2.5}$ and PM$_{10}$ from the US EPA Air Quality System (AQS; https://www.epa.gov/aqs, last access: 3 March 2022). IMPROVE is a collaborative association of state, tribal, and federal agencies, and international partners. US Environmental Protection Agency is the primary funding source, with contracting and research support from the National Park Service. The Air Quality Group at the University of California, Davis is the central analytical laboratory, with ion analysis provided by Research Triangle Institute, and carbon analysis provided by Desert Research Institute. We convert observed organic carbon to organic matter assuming a mass-to-carbon ratio of 1.8, we calculate a concentration of mineral dust as 2.49 Si + 2.20 Al + 1.63 Ca + 2.42 Fe + 1.94 Ti, and we calculate a concentration of sea salt as 3.25 Na for comparison to the model results.

We evaluate the SRIM and 1L2B simulations against the IMPROVE, CSN, and AQS data by calculating the correlation coefficient (R), the normalized Mean Bias (NMB), the Root Mean Square Error (RMSE), and the fraction of simulated data within a factor of 2 of the observations (Fac2). As noted previously, the concentrations of aerosol species in the HYGRO and 1L2B simulations are similar. We note that R, NMB, and Fac2 differed by ≤0.03 and RMSE differed by <0.01 $\mu$g m$^{-3}$ between the SRIM and 1L2B results for sulphate, nitrate and ammonium from the IMPROVE and CSN networks. The SRIM and 1L2B simulations therefore compare similarly well to observations for these species. This is expected, as these species are more weakly affected by the difference in mixing-state representation. There is an existing high bias in the SRIM-predicted





concentrations of BC, organic aerosol, and dust. This high bias is worsened in the 1L2B simulation, due to the slower removal of these species by wet deposition in the 1L2B simulation, and this affects the calculated NMB, RMSE, and Fac2 values for these species. The correlation coefficients for EC and organic aerosol are not strongly affected. This suggests that the variability in BC and organic aerosol concentrations is not primarily controlled by wet deposition at these sites during the case study time period. As discussed, the wet deposition of dust is likely reduced too much in the 1L2B simulation, which may be

responsible for the lower correlation between the observed and simulated dust concentrations in the 1L2B simulation. There is a slight shift of the sea salt size distribution to larger sizes in the 1L2B simulation, perhaps due to more coagulation with the larger concentrations EC, organic aerosol, and dust. This reduces the fine sea salt aerosol mass, even while total sea salt aerosol concentrations slightly increase. The NMB and RMSE for sea salt is therefore reduced in the 1L2B simulation compared to the SRIM simulation. The increased concentrations of $PM_{2.5}$ and $PM_{10}$ in the 1L2B simulation increase the already high bias

in $PM_{2.5}$ and reduces the underprediction of $PM_{10}$, as compared to the SRIM simulation. However, in both cases the RMSE is reduced, and the R and Fac2 values are either unchanged or slightly improved.

### 3.1.3   Air Quality Health Index

The Air Quality Health Index (AQHI; Stieb et al., 2008) is used by Environment and Climate Change Canada to communicate

adverse health risks due to poor air quality to Canadians. It is formulated as a scale that ranges from 0 (excellent air quality) to 10 (very poor air quality), and is calculated based on the concentrations of $PM_{2.5}$ or $PM_{10}$, ozone ($O_3$), and nitrogen dioxide ($NO_2$). While the equations for calculating the AQHI permit values greater than 10 under exceptionally high concentrations of $PM_{2.5}$, $PM_{10}$, $O_3$, or $NO_2$, we restrict the values of AQHI to a maximum of 10, both because this is the intended range of the AQHI, and to reduce the influence of exceptional, highly concentrated plumes in uninhabited areas (such as those from forest

fires) on our results.

The concentration of $O_3$ was, on average, 0.05% less in the HYGRO or 1L2B cases than the SRIM case, and the concentration of $NO_2$ was, on average, 0.2% greater in the HYGRO or 1L2B cases than the SRIM case. We can therefore attribute differences in the AQHI primarily to differences in $PM_{2.5}$ and $PM_{10}$. The $PM_{2.5}$ AQHI was, on average, 0.04 units greater in the HYGRO

and 1L2B simulations than in the SRIM simulation, and the $PM_{10}$ AQHI was 0.06 units greater in the HYGRO and 1L2B simulations than in the SRIM simulation. This can be seen in the spatial patterns of the differences in AQHI, shown in Fig. 3 for $PM_{2.5}$ AQHI and in Fig. S2 for $PM_{10}$ AQHI.

### 3.1.4   BC mass fraction in BC-containing particles

Before discussing the BC mass fractions, we will discuss the concentrations of BC in more detail. We show in Fig. 4 the mean

BC concentrations at the surface from the SRIM simulation, as well as the absolute and relative differences in the BC mixing ratio between the HYGRO and SRIM simulations. We note that during the time period simulated, several large forest fires burned in Alaska and northern Canada, and the influence of these fires on $PM_{2.5}$, $AQHI_{2.5}$, and BC concentrations are clearly



**Table 3.** Evaluation of SRIM and 1L2B simulations against observations. N = number of model/observation pairs, R = correlation coefficient, NMB = normalized Mean Bias, RMSE = Root Mean Square Error, Fac2 = fraction within a factor of two. Better performance between SRIM and 1L2B (larger values of R and Fac2 and smaller values of NMB and RMSE) are in bold font.

| | | N | R | NMB | RMSE | Fac2 |
|---|---|---|---|---|---|---|
| IMPROVE Daily Fine Sulfate [$\mu$g m$^{-3}$] | SRIM | 1543 | 0.38 | -0.10 | 0.88 | 0.72 |
| | 1L2B | 1543 | **0.39** | **-0.09** | 0.88 | 0.72 |
| CSN Daily Fine Sulfate [$\mu$g m$^{-3}$] | SRIM | 1050 | 0.28 | -0.22 | 1.24 | 0.69 |
| | 1L2B | 1050 | 0.28 | **-0.21** | 1.24 | **0.70** |
| IMPROVE Daily Fine Nitrate [$\mu$g m$^{-3}$] | SRIM | 1543 | 0.79 | -0.72 | 0.34 | 0.10 |
| | 1L2B | 1543 | 0.79 | **-0.71** | 0.34 | 0.10 |
| CSN Daily Fine Nitrate [$\mu$g m$^{-3}$] | SRIM | 1045 | 0.52 | -0.20 | 0.96 | 0.20 |
| | 1L2B | 1045 | 0.52 | **-0.19** | 0.96 | **0.21** |
| CSN Daily Fine Ammonium [$\mu$g m$^{-3}$] | SRIM | 1005 | 0.41 | **1.08** | 0.48 | 0.31 |
| | 1L2B | 1005 | 0.41 | 1.11 | 0.48 | 0.31 |
| IMPROVE Daily Fine EC [$\mu$g m$^{-3}$] | SRIM | 1573 | 0.26 | **0.64** | 0.51 | **0.57** |
| | 1L2B | 1573 | 0.26 | 0.74 | 0.51 | 0.54 |
| CSN Daily Fine EC [$\mu$g m$^{-3}$] | SRIM | 956 | 0.34 | **0.27** | **0.62** | 0.70 |
| | 1L2B | 956 | 0.34 | 0.31 | 0.63 | **0.71** |
| IMPROVE Daily Fine Organic Matter [$\mu$g m$^{-3}$] | SRIM | 1573 | 0.26 | **1.01** | **5.73** | **0.52** |
| | 1L2B | 1573 | 0.26 | 1.19 | 5.94 | 0.44 |
| CSN Daily Fine Organic Matter [$\mu$g m$^{-3}$] | SRIM | 956 | **0.33** | **1.03** | **7.04** | **0.56** |
| | 1L2B | 956 | 0.32 | 1.19 | 7.42 | 0.50 |
| IMPROVE Daily Fine Dust [$\mu$g m$^{-3}$] | SRIM | 1539 | **0.64** | **0.76** | **2.15** | **0.39** |
| | 1L2B | 1539 | 0.61 | 1.50 | 3.21 | 0.32 |
| CSN Daily Fine Dust [$\mu$g m$^{-3}$] | SRIM | 1060 | **0.68** | **1.71** | **3.18** | **0.29** |
| | 1L2B | 1060 | 0.64 | 2.59 | 4.53 | 0.24 |
| IMPROVE Daily Fine Sea Salt [$\mu$g m$^{-3}$] | SRIM | 1534 | 0.63 | 4.18 | 2.16 | 0.10 |
| | 1L2B | 1534 | **0.64** | **4.04** | **2.10** | **0.11** |
| CSN Daily Fine Sea Salt [$\mu$g m$^{-3}$] | SRIM | 1007 | 0.64 | 4.51 | 2.13 | 0.12 |
| | 1L2B | 1007 | 0.64 | **4.45** | **2.08** | 0.12 |
| AQS Hourly PM2.5 [$\mu$g m$^{-3}$] | SRIM | 290614 | 0.06 | **0.54** | 67.45 | 0.53 |
| | 1L2B | 290614 | **0.07** | 0.70 | **66.86** | 0.53 |
| AQS Hourly PM10 [$\mu$g m$^{-3}$] | SRIM | 238699 | 0.02 | -0.40 | 97.98 | 0.44 |
| | 1L2B | 238699 | 0.02 | **-0.31** | **97.96** | **0.45** |





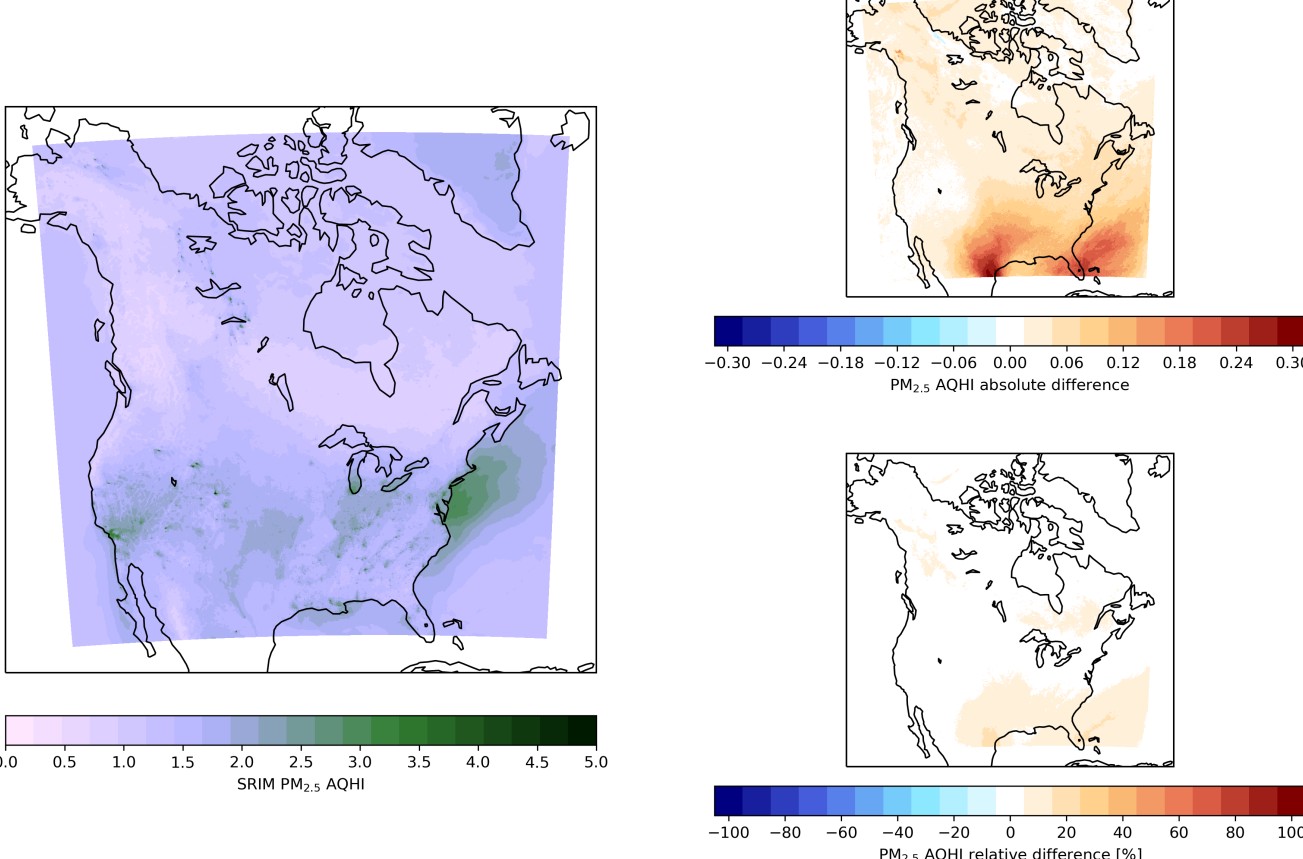

**Figure 3.** left: mean PM$_{2.5}$ AQHI values from the SRIM simulation; top right: mean difference in PM$_{2.5}$ AQHI values between the HYGRO and SRIM simulations; bottom right: relative difference in mean PM$_{2.5}$ AQHI values between the HYGRO and SRIM simulations. Note that PM$_{2.5}$ AQHI values are nearly identical in the HYGRO and 1L2B simulations.

visible in Figures 1, 3, and 4, respectively.

As discussed in Sect. 3.1.1, the concentrations of BC typically increase in the HYGRO and 1L2B simulations, as less BC is removed through wet deposition. However, there are notable locations downwind of forest fires in Alaska and northern Canada where the concentrations of BC at the surface decrease. Aerosol in the high-$\kappa$ category and all aerosol in the SRIM simulation that is sufficiently large can be ingested into cloud droplets. These cloud droplets can grow to drizzle sizes, and would then be subject to gravitational settling. If the drizzle droplets evaporate before reaching the surface, they will transport any aerosol
mass in the droplets to lower altitudes. However, aerosol in the low-$\kappa$ categories in the HYGRO and 1L2B simulations are not subject to this process, and therefore BC that is lofted to higher altitudes takes longer to reach the surface in these simulations,



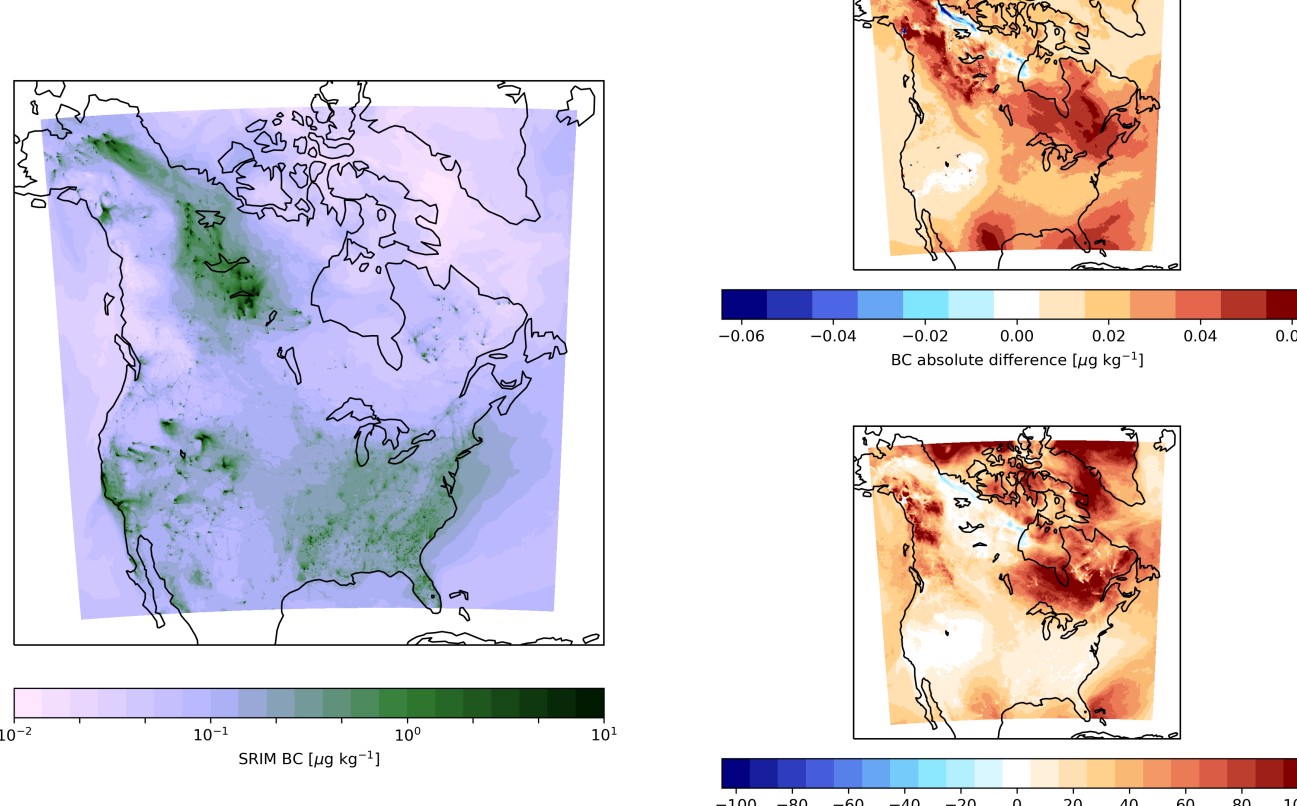

**Figure 4.** left: mean BC concentrations from the SRIM simulation; top right: mean difference in BC concentrations between the HYGRO and SRIM simulations; bottom right: relative difference in mean BC concentrations between the HYGRO and SRIM simulations. Note that BC concentrations are nearly identical in the HYGRO and 1L2B simulations.

reducing the surface concentrations close to the forest fires and increasing them further downwind.

In order to explain the effects of the mixing-state representation on aerosol-radiation interactions (discussed further in
Sect. 3.1.5), we calculate the BC mass fraction within particles that contain BC. To do this, we calculate the BC mass fraction in each combination of size bin and mixing-state category separately, then report the average value weighted by the mass of BC in the same combination of size bin and mixing-state category. We show these results in Fig. 5. The SRIM configuration assumes that BC is internally mixed with all other aerosol mass in particles of the same size, thus the BC mass fraction is similar to the total BC mass divided by the total aerosol mass. For 99% of the grid cells, the mean BC mass fraction in the
SRIM simulation is less than 5%. The HYGRO configuration shows modest improvements over the SRIM configuration, while





the 1L2B configuration is able to resolve many regions close to emission sources where BC is thinly coated (high BC mass fractions).

We note that the 1L2B simulation is better able to capture regions with large BC mass fractions than the HYGRO simulation
because the HYGRO configuration assumes that all low-hygroscopicity species within the same size bin are internally mixed, including BC, dust, and POA. Most dust mass exists in larger size bins than BC. Therefore, even the SRIM simulation does not assume much internal mixing of BC and dust. However, BC and POA are emitted into the same size bins and from the same source regions. When the BC is assumed to be internally mixed with other low-hygroscopicity species, the resulting particles frequently consist of BC thickly coated with POA. The 1L2B simulation is able to distinguish BC thinly coated with POA
from BC thickly coated with POA, and it predicts that a large proportion of BC near source regions has only a thin coating of non-BC species.

### 3.1.5   Aerosol-radiation interactions

We show the monthly mean AOD from the SRIM simulation and the difference between the HYGRO and SRIM simulations in
Fig. 6. We remind the reader that the calculations of aerosol optical properties are restricted to daylight hours in GEM-MACH. As such, we include only data from between 1300 and 2100 UTC in Fig. 6, in order to exclude times of day when the AOD was not calculated for some part of the domain shown. We also note that the mean AOD in the 1L2B and HYGRO simulations differs by no more than 0.0011 for any grid cell in the domain. When using the HYGRO configuration, the AOD is 34% larger than in the SRIM case. A comparison of the AOD with the absorption aerosol optical depth (AAOD) (see Table 1 and Fig. 7)
reveals that the AOD is dominated by aerosol scattering, rather than aerosol absorption. Previous studies have found that the optical properties of non-absorbing aerosol is not strongly sensitive to the mixing-state of the aerosol (e.g. Zaveri et al., 2010; Klingmüller et al., 2014), and that because AOD is dominated by the scattering component, ambient AOD is not strongly sensitive to mixing-state (e.g. Matsui et al., 2013, 2014; Klingmüller et al., 2014; Han et al., 2013), although a recent study has shown that aerosol scattering can be very sensitive to aerosol mixing-state under certain conditions (Yao et al., 2022). We
therefore do not expect our more-detailed representation of the BC mass to yield strong changes in aerosol scattering, but we do expect a decrease in aerosol absorption. We therefore conclude that the differences are due predominantly to the increases in aerosol mass, in turn due to the decrease in aerosol wet deposition. This is supported by the fact that the aerosol AOD and differences in AOD are visibly well-correlated with $PM_{2.5}$ and the differences in $PM_{2.5}$ shown in Fig. 1.

We show the monthly mean AAOD from the SRIM simulation and the differences between the 1L2B, HYGRO and SRIM simulations in Fig. 7. The AAOD is 39% higher in the HYGRO case than in the SRIM case. As shown in Fig. 5, the BC mass fraction in BC-containing particles is only slightly larger in the HYGRO case than the SRIM cases. If the mass concentrations of all aerosol species were equal in both cases, higher BC mass fractions would imply thinner coatings and smaller absorption enhancements for the BC-containing particles. This effect would be expected to reduce the AAOD in the HYGRO case as





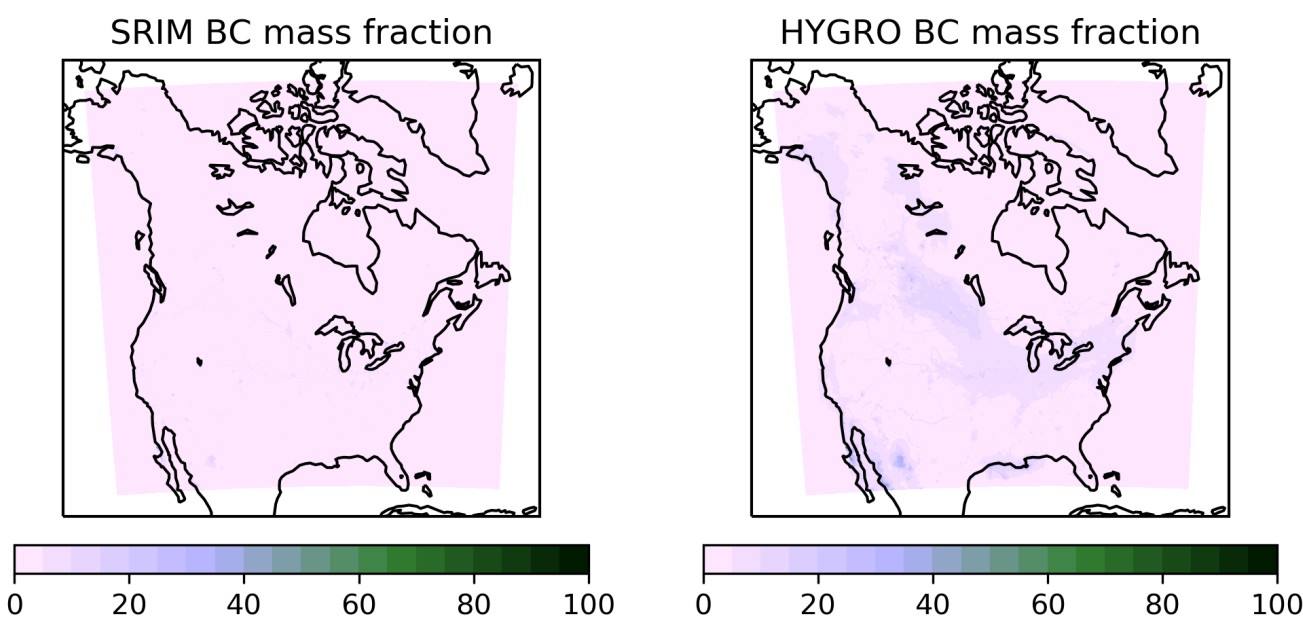

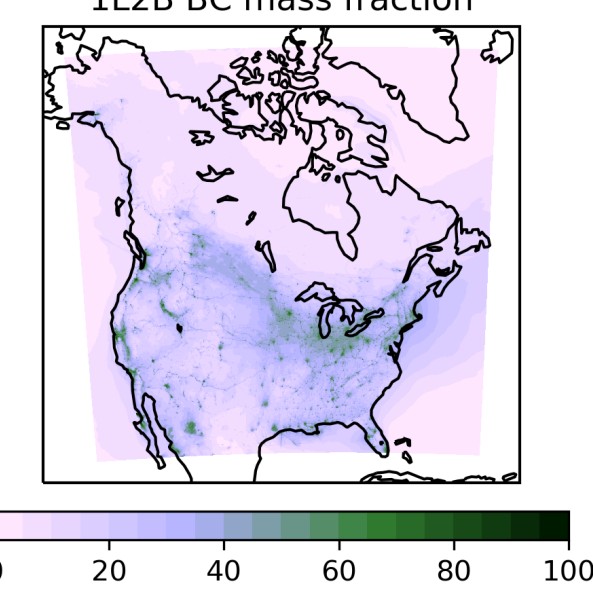

**Figure 5.** Black carbon mass fractions in BC-containing particles. Values are given at the surface, and weighted by the BC mass in each size bin and mixing-state category. Values are shown for the following simulations: top left: SRIM; top right: HYGRO; Bottom left: 1L2B.



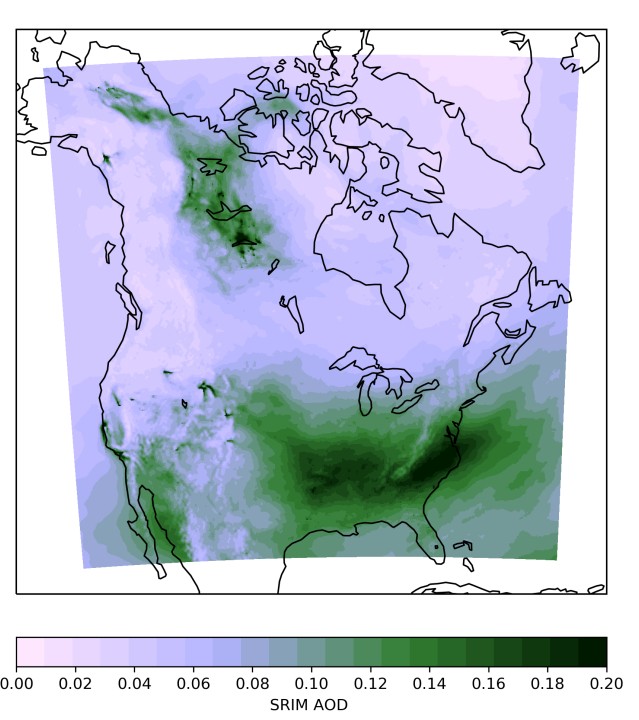

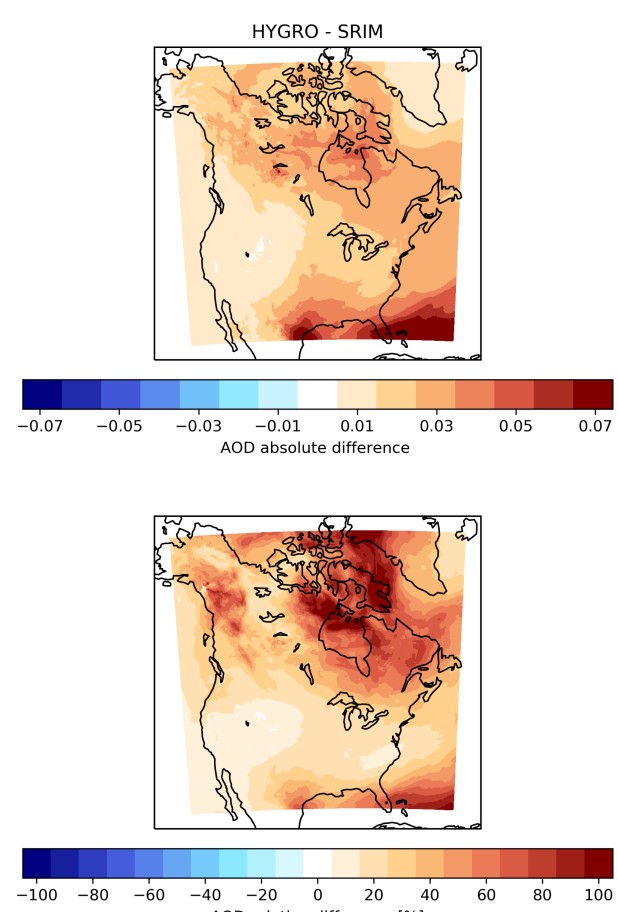

**Figure 6.** left: mean AOD from the SRIM simulation; top right: mean difference in AOD between the HYGRO and SRIM simulations; bottom right: relative difference in mean AOD between the HYGRO and SRIM simulations. Only results from the hours of 1300-2100 UTC are included as the AOD is only calculated during local daylight hours.

compared to the SRIM case. The simulated increase in AAOD is due primarily to the increased concentrations of BC in the HYGRO case compared to the SRIM case.

When using the 1L2B configuration, the AAOD is on average 3% less than in the HYGRO case, with these decreases being primarily over the eastern United States and around the Gulf of California. These are similar to the regions where the 1L2B case

has higher BC mass fractions than the HYGRO case, as shown in Fig. 5, and are typically downwind of large anthropogenic sources of BC. We note that there are smaller differences in the plumes of the large northern forest fires; this is because emissions of BC from forest fires are assumed to be thickly coated as observations (Perring et al., 2017; Kondo et al., 2011) have shown that this is typically the case. Around the Gulf of Mexico and just south the Great Lakes region, the decreases





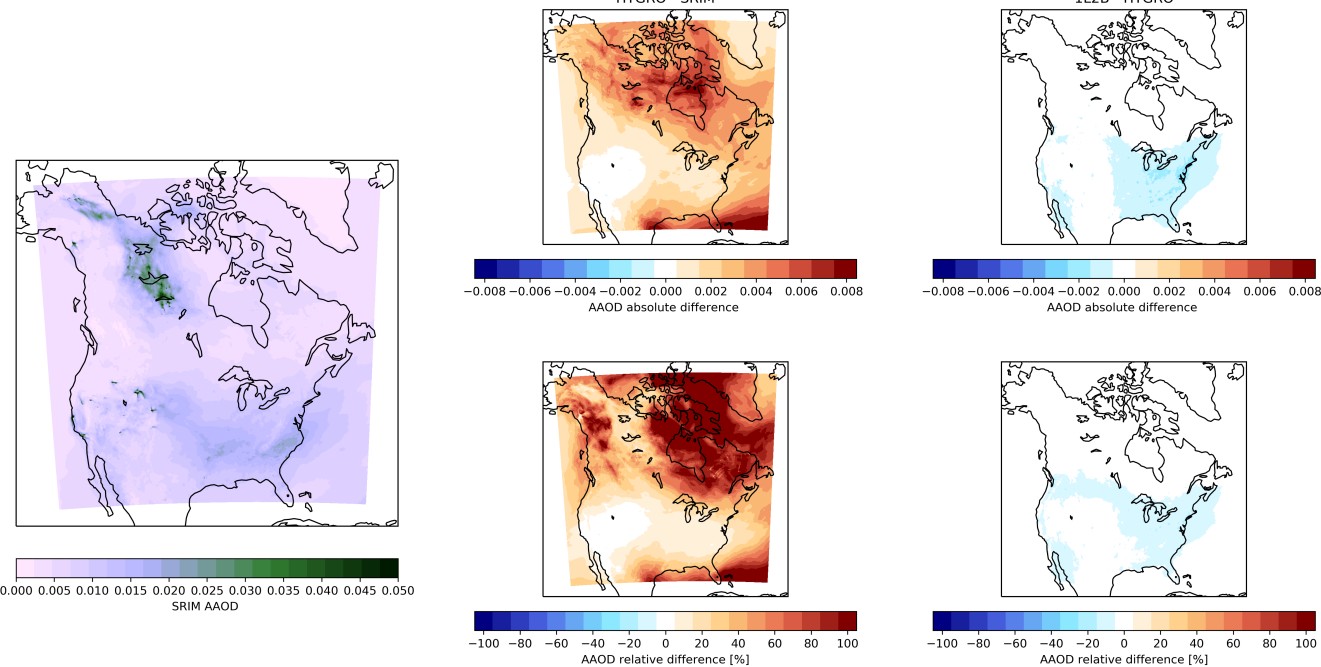

**Figure 7.** left: mean AAOD from the SRIM simulation; top centre: mean difference in AAOD between the HYGRO and SRIM simulations; top right: mean difference in AAOD between the 1L2B and HYGRO simulations; bottom centre: relative difference in mean AAOD between the HYGRO and SRIM simulations; bottom right: relative difference in mean AAOD between the 1L2B and HYGRO simulations. Only results from the hours of 1300-2100 UTC are included as the AOD is only calculated during local daylight hours.

in AAOD between 1L2B and HYGRO (due to better-resolving the coating thickness on BC) are larger in magnitude than

the increases in AAOD between HYGRO and SRIM (due to reduced wet deposition of low-hygroscopicity aerosol, including BC). Therefore in these regions, there is a net decrease in AAOD from the SRIM to 1L2B, when both the effects of mixing-state on wet deposition and absorption enhancement is considered, while for most of the rest of the North American domain, AAOD increases as the effect due to decreases in wet deposition is more important that the effect due to decreased absorption enhancement.

**3.2   Aerosol-Meteorology Feedbacks**

In order to examine the interactions between aerosol mixing-state representation and meteorology, we will now describe the results of the aerosol-meteorology feedbacks simulations. In these simulations, the cloud droplet number concentration is parameterized based on the aerosol size distribution using Abdul-Razzak (2002), as described in Sect. 2. In the case of multiple mixing-state categories, the distinct composition of aerosol in each mixing-state category is considered, so aerosol in different

mixing-state categories will have different critical radii for activation under the same atmospheric conditions. Additionally, aerosol and trace gas concentrations are permitted to reduce incoming radiation, which would subsequently alter atmospheric





and surface energy balances.

As our focus is on the effects of differences in the aerosol mixing-state representation, we only compare cases with aerosol-
meteorology feedbacks to other cases with aerosol-meteorology feedbacks. For comparisons of GEM-MACH results with and
without aerosol-meteorology feedbacks, we refer the reader to Gong et al. (2015) and Makar et al. (2015a, b).

### 3.2.1    Aerosol-cloud interactions

In order to target low clouds most likely to be affected by aerosol emitted from the surface, we restrict our analysis to the
clouds with model hybrid levels between 0.807 and 0.962, approximately 35-185 hPa below surface pressure. As all cloud
variables were saved as 3-hourly means, which will include transitions between cloudy and cloud-free periods, our reported
cloud properties will have smaller values than if we had analyzed instantaneous model output. This includes, most notably, the
cloud droplet and raindrop number mixing ratios. However, as our interest is in the comparison between simulations, which are
all treated identically, this would not alter our conclusions. Additionally, in order to provide more physically meaningful values,
when calculating temporally and horizontally averaged cloud properties we define "cloudy" grid cells as those with 3-hourly
cloud water mixing ratios ($Q_C$) >0.005g kg$^{-1}$, and we filter out grid cells with lower 3-hourly $Q_C$ values. The mean number of
cloudy grid cells differs by less than 0.7% between simulations (not shown), with the HYGRO_feedbacks and 1L2B_feedbacks
simulations having slightly more cloudy grid cells than the SRIM_feedbacks simulation. Therefore differences between sim-
ulations are better explained as changes in in-cloud properties, rather than as changes in the spatial extent of clouds. We note
that the cloud fraction over the western United States was low during July of 2016, as evidenced in the MODIS satellite
retrievals (NASA Earth Observations, https://neo.gsfc.nasa.gov/view.php?datasetId=MODAL2_M_CLD_FR&date=2016-07-
28, last access November 19th, 2021).

We show in Fig. 8 the vertical distributions of the temporally and horizontally meaned in-cloud cloud droplet number mix-
ing ratios ($N_C$), cloud water mixing ratios ($Q_C$), rain drop number mixing ratios ($N_R$), and rain water mixing ratios ($Q_R$) in
the SRIM_feedbacks, HYGRO_feedbacks, and 1L2B_feedbacks simulations. The HYGRO_feedbacks and 1L2B_feedbacks
model simulations predict $N_C$ values that are approximately 15% larger than in the SRIM_feedbacks simulation. The difference
in $N_C$ is approximately constant with altitude. This difference is due to increased aerosol number concentrations, in turn due to
both greater aerosol mass concentrations and smaller aerosol diameter, as shown in Fig. 9. These increased $N_C$ values lead to
mean $Q_C$ values that are about 7% greater than in the SRIM_feedbacks simulation. As $N_C$ increases more than $Q_C$, the mean
cloud droplet size will be decreased in the HYGRO_feedbacks and 1L2B_feedbacks simulations. These reduced cloud droplet
sizes would be expected to result in reduced autoconversion and slower drizzle formation. Indeed, both $N_R$ and $Q_R$ are reduced
in the HYGRO_feedbacks and 1L2B_feedbacks simulations relative to the SRIM_feedbacks simulation, by about 20% for $N_R$
and 9% for $Q_R$. The difference in $N_C$ is approximately constant with altitude, while the differences in $Q_C$, $N_R$ and $Q_R$ increase
with altitude. For all cloud variables, the differences are slightly larger in the 1L2B_feedbacks simulation compared to the





HYGRO_feedbacks simulation.

The decreases in in-cloud $Q_R$ discussed above would be expected to result in decreases in precipitation at the surface. We show the mean precipitation from the SRIM simulation and the effects on precipitation of the HYGRO and 1L2B mixing-state representations in Fig. 10. Many of the differences shown in Fig. 10 include large decreases near large increases. These are due

in part to small changes in advection patterns, which subsequently alter the locations of precipitation. We can determine the net effect of the difference in mixing-state representation on surface precipitation by averaging across the domain. When spatially and temporally averaged, the effects mixing-state representation on precipitation is modest: In the HYGRO_feedbacks and 1L2B_feedbacks simulations, the precipitation is reduced by 0.6% relative to the SRIM_feedbacks simulation, much smaller than the differences in in-cloud $Q_R$ discussed above. As the decreases in $N_R$ are greater than those in $Q_R$, the HY-

GRO_feedbacks and 1L2B_feedbacks simulations would have larger rain drops than the SRIM_feedbacks simulation, and these larger rain drops would settle to the surface more efficiently, thereby partially offsetting the reduction in $Q_R$.

The increases in $N_C$ and $Q_C$ shown above, along with the small increases in AOD shown in Sect. 3.1.5, would be expected to reduce the shortwave radiation reaching the surface and to potentially reduce surface temperatures. We show in Fig. 11

the differences in mean surface temperatures between the HYGRO_feedbacks and SRIM_feedbacks simulations and between the 1L2B_feedbacks and HYGRO_feedbacks simulations. Between HYGRO_feedbacks and SRIM_feedbacks, eastern and southern North America shows either small differences or noisy differences that would be consistent with slight changes in the locations of clouds. However, there is a clear increase of about 0.01 K over large areas of the oceans and a clear decrease of about 0.06 K over northern Quebec and eastern Nunavut. We note that this region encompasses the outflow of forest fires that

occurred in north-eastern Canada during the simulation, as is visible in the differences in surface BC concentrations (Fig. 4). In the HYGRO_feedbacks simulation, the emissions from these forest fires are removed more slowly by wet deposition. Therefore, more aerosol particles remain to act as CCN further downwind from the source. The greater CCN concentration increases both $N_C$ and $Q_C$ within the cloud, reducing the solar radiation reaching the surface, and reducing surface temperatures. The differences in surface temperatures between 1L2B_feedbacks and HYGRO_feedbacks are noisy throughout the domain, con-

sistent with slight changes in the locations of clouds. We therefore cannot determine any clear effect on surface temperatures due to differences in mixing-state representation between these two simulations.

We note that for all cloud properties, there are only small differences between 1L2B_feedbacks and HYGRO_feedbacks. We remind the reader that the differences in mixing-state representation between 1L2B and HYGRO were designed to capture

the effects of correctly resolving the thickness of non-absorbing shells on BC and the subsequent enhancement in aerosol absorption. The effects of these differences in absorption enhancement would be permitted to affect atmospheric temperatures in our simulations, with potential subsequent effects on atmospheric stability. However, we do not find a strong effect on cloud properties, surface precipitation or surface temperatures in this study. This may be, in part, due to our choice of case study.





**Figure 8.** Vertical distribution of temporally and horizontally averaged cloud droplet number mixing ratios ($N_C$, a), cloud water mixing ratios ($Q_C$, b), rain drop number mixing ratios ($N_R$, c), and rain water mixing ratios ($Q_R$, d) in the SRIM_feedbacks, HYGRO_feedbacks, and 1L2B_feedbacks simulations. The shading indicates one temporal standard deviation about the mean.





**Figure 9.** Vertical distribution of temporally and horizontally averaged aerosol properties in cloudy grid cells in the SRIM_feedbacks, HYGRO_feedbacks, and 1L2B_feedbacks simulations. Left: total mass mixing ratios. Right: mass-mean aerosol diameter. The shading indicates one temporal standard deviation about the mean.





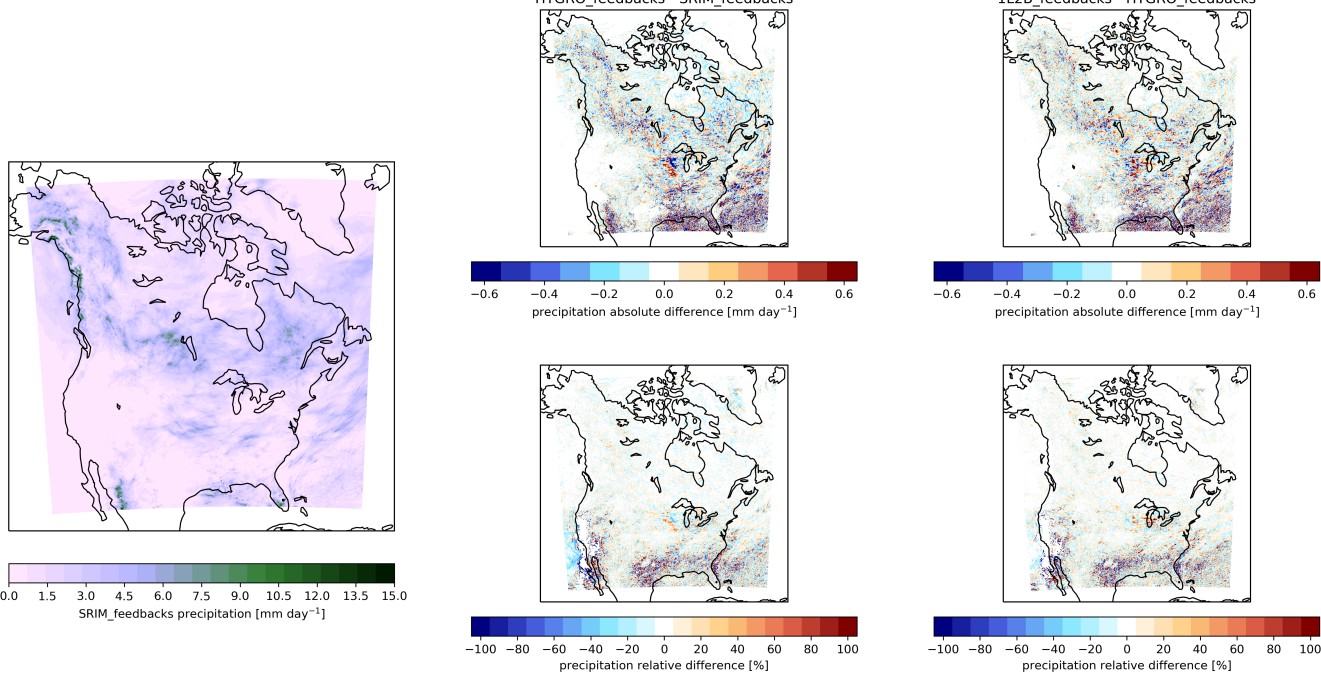

**Figure 10.** Left: mean precipitation from the SRIM_feedbacks simulation; top centre: mean difference in precipitation between the HY-GRO_feedbacks and SRIM_feedbacks simulations; top right: mean difference in precipitation between the 1L2B_feedbacks and HY-GRO_feedbacks simulations; bottom centre: relative difference in mean precipitation between the HYGRO_feedbacks and SRIM_feedbacks simulations; bottom right: relative difference in mean precipitation between the 1L2B_feedbacks and HYGRO_feedbacks simulations.

## 4 Conclusions

In this study, we have implemented a detailed representation of aerosol mixing-state into the GEM-MACH air quality and weather forecast model. Our mixing-state representation includes three categories: one for more-hygroscopic aerosol, one for less-hygroscopic aerosol with a high BC mass fraction, and one for less-hygroscopic aerosol with a low BC mass fraction. This is the first model with a mixing-state representation of this type simulating a continent-scale domain. Currently, the HYGRO and 1L2B configurations require approximately 70% and 150% more running-time, respectively, than the SRIM configuration. We expect to reduce this additional cost through improvements to the efficiency of the model tracer transport scheme in the near future. The more-detailed representation allowed us to better resolve two different aspects of aerosol mixing state: First, differences in hygroscopicity due to differences in aerosol composition, including the change in hygroscopicity with time as less-hygroscopic aerosol becomes coated with hydrophilic material. Second, the thickness of non-absorbing coatings on BC aerosol which enhance the absorption of the BC aerosol.





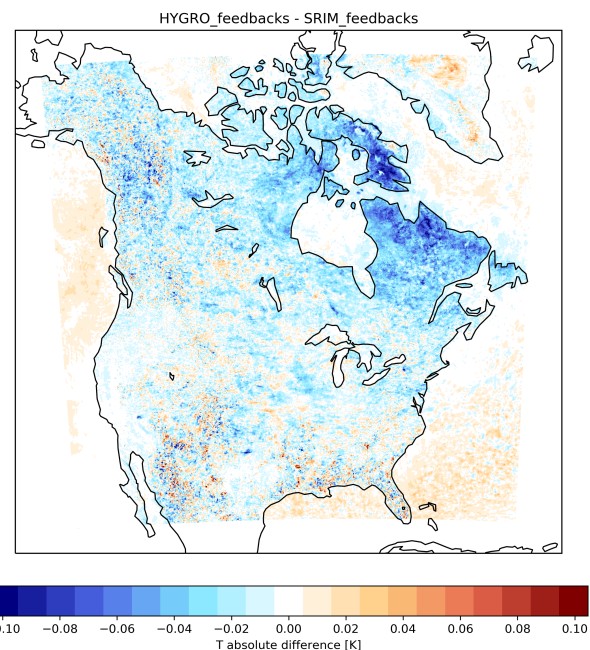
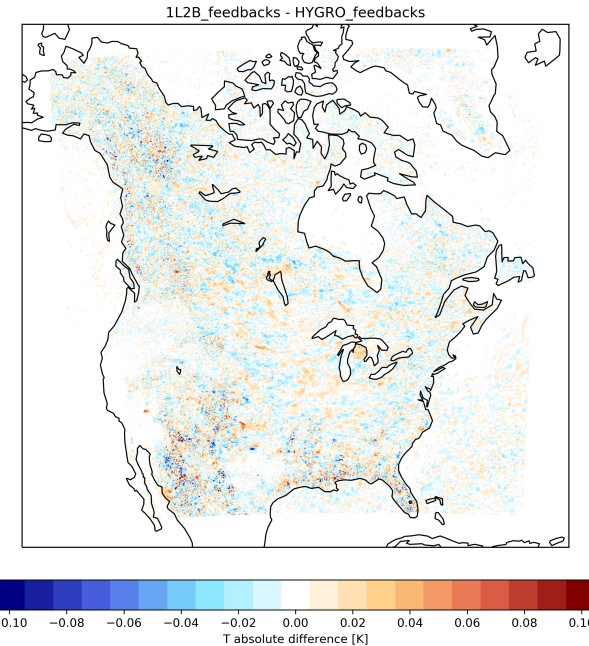

**Figure 11.** Left: mean difference in surface temperature between the HYGRO_feedbacks and SRIM_feedbacks simulations; right: mean difference in surface temperature between the 1L2B_feedbacks and HYGRO_feedbacks simulations.

We compared the results of the three-category representation (1L2B) with a simulation that uses two categories, split by hygroscopicity (HYGRO), and a simulation using the original size-resolved internally-mixed assumption (SRIM). We showed that when we included one or two categories of less-hygroscopic aerosol, wet deposition of BC, POA, SOA and dust was

reduced, yielding increases in the mean concentrations of these species of 16-93%, and an increase in the mean $PM_{2.5}$ concentration by 23%. The effect on dust concentrations is likely overestimated, as the current implementation prevents wet deposition of aerosol in the hydrophobic category, even if the aerosol is large. We intend to improve on this in a future version of GEM-MACH. As BC, POA, and SOA mass is more concentrated in smaller aerosol particles, we believe that the reductions of wet deposition in these species is realistic. The increased $PM_{2.5}$ concentrations led to an increase in the $AQHI_{2.5}$ by 0.05 units on

average. The increases in aerosol concentrations also led to increases in both AOD and AAOD.

We briefly compared the results of the SRIM and 1L2B simulations and observations from the IMPROVE, CSN and AQS networks. However, we did not find significant improvement in model-observation agreement with the more-detailed mixing-state representation. The reduced wet deposition worsened an existing high bias in BC, organic matter, and dust concentrations,

and we saw only small changes in correlation with the observations. It is likely that a more thorough assessment will require observations from sites that are strongly affected by long-range transport of BC and organic aerosol. The CSN network sites in particular are located in urban centres, and would therefore be expected to be weakly affected by changes in wet deposition.



We will investigate this further in future work.

However, using two categories to resolve more-hygroscopic and less-hygroscopic aerosol only yielded modest improvements in resolving the amount of coating material on BC particles, which alters their absorption of solar radiation. We found that using three mixing-state categories (more-hygroscopic, less-hygroscopic high BC mass fraction, less-hygroscopic low BC mass fraction) allowed us to distinguish thinly coated BC from BC that was thickly coated with POA. This yielded a mean AAOD that was 3% less than when separating the aerosol by hygroscopicity alone. Many sources of BC are also sources of POA, and observations indicate that the BC-containing particles frequently also contain POA, even close to emission sources (Perring et al., 2017; Kondo et al., 2011). We note that we assumed that particles from area sources were externally mixed at emission. This assumption will yield a maximum difference between our sensitivity simulations. Nonetheless, as thinly coated BC particles have been observed in the ambient atmosphere, even far from emission sources (Zanatta et al., 2018; Sharma et al., 2017), it is clear that POA and BC are not evenly distributed across particles in the same size range. The proportion of POA that is emitted as BC-containing particles vs. BC-free particles is currently poorly constrained. We therefore suggest that future observation campaigns record not only the coating thickness on BC-containing particles, but also, when possible, the proportion of organic matter that exists as BC-free particles vs. BC-containing particles.

We then performed simulations that included aerosol feedbacks on meteorology in order to determine the effects of mixing-state representation on the forecast meteorology. We found a clear effect due to including two categories of aerosol hygroscopicity: the increased aerosol concentrations due to the decreases in wet deposition increased cloud droplet mixing ratios by approximately 15%. This led to a reduction in the mean precipitation by 0.6%. The increased cloud reflectivity resulted in a decrease in surface temperatures by about 0.06 K over northeastern Canada, in the outflow of large forest fires. When we compared the results of the HYGRO simulation with those of the IL2B simulation, which better resolves BC mass fraction and aerosol absorption, we did not find a strong effect on forecast meteorology.

*Code availability.* GEM-MACH, the atmospheric chemistry library for the GEM numerical atmospheric model (© 2007–2013, Air Quality Research Division and National Prediction Operations Division, Environment and Climate Change Canada), is a free software which can be redistributed and/or modified under the terms of the GNU Lesser General Public License as published by the Free Software Foundation – either version 2.1 of the license or any later version. The GEM (meteorology) code (CMC, 2021) is available to download from https://github.com/mfvalin?tab=repositories (last access: 27 April 2022). The executable for GEM-MACH is obtained by providing the chemistry library to GEM when generating its executable.



*Author contributions.* RS designed and implemented the novel mixing-state representations in GEM-MACH. RS led the writing and preparation of the manuscript. RS and AR performed the computational experiments and led the analysis. MM designed and implemented improved

aerosol-radiation interactions in GEM-MACH. AD provided project oversight and guidance.

*Competing interests.* We declare that no competing interests are present.

*Acknowledgements.* We would like to thank Ayodeji Akingunola, Alexandru Lupu and Craig Stroud of Air Quality Research Division of Environment and Climate Change Canada for their expertise and contributions in this projects.



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
