# Peer review of "An Improved Representation of Aerosol Mixing State for Air-Quality-Weather Interactions"

_EGUsphere, 2022_

## Referee Comment (RC1)

An Improved Representation of Aerosol Mixing State for Air-Quality-Weather Interactions
Stevens et al., 2022

Summary

This study is about a development of a model representation of the aerosol mixing state in a chemical transport model called GEM-MACH. The authors developed a three-category representation (1L2B) to resolve an aerosol population into three categories, i.e. (1) high hygroscopicity, (2) low hygroscopicity with high BC content, and (3) low hygroscopicity with low BC content. The authors further compared the 1L2B representation to the other two representations called HYGRO (2-category, high hygroscopicity, low hygroscopicity) and SRIM (the original one in GEM-MACH which assume complete internally-mixed aerosols) by evaluating against the observation of a case of biomass-burning over North America. Besides, the study investigated the impact of aerosol mixing state representation on meteorology with and without feedback. The study matches the scope of *ACP*. I recommend acceptance of this manuscript for publication upon satisfactorily clarifications of the following issues.

Questions/clarifications

1. (Line 25) Apart from condensation and coagulation, chemical reactions could also modify aerosol mixing state (a.k.a. chemical aging of aerosol). I suggest the authors to clarify this.

2. (Line 69-75) In the aerosol representation of 1L2B, the fixed bin boundary of hygroscopicity (kappa) and BC mass fraction of 0.1 and 0.3 proposed by Ching et al., 2016 was derived based on urban environments. It is possible that in the scenarios of biomass burning (test case in this study), such fixed boundary values may not be the most optimal in resolving aerosol mixing state. Should the author provide some sensitivities studies on the bin boundary positions, it would enhance the scientific values of this manuscript and applicability of the model representation to various kinds of environments.

3. (Section 2.1) Apart from the atmospheric physical and chemical processes, the sensitivity study of aerosol mixing state also depends on mixing state of the aerosols at the instant of their

emissions. It is expected that the authors should state clearly or tabulate the assumed chemical composition of the particles from biomass burning (i.e. smoke particles from forest fire) as well as other aerosol types when they are emitted. This would definitely affect the evaluation of the simulations.

4. (Line 303) According to the latest particle-resolved study published by *Yao et al., 2022 ACP*, based on 1800 environmental scenarios, simplified aerosol representation (assuming internally-mixed aerosol particles) could potentially lead to a wide range of underestimation in aerosol scattering with a maximum of -32% among the environment scenarios they studied. Therefore, I think it is better for the authors to clarify in the text that error in scattering coefficient of an aerosol population may not be negligible, instead it could depend on the interaction between the aerosol particles and the environment, such as relative humidity and chemistry processes involved.

5. (Line 311) Please explain why "higher black carbon mass fraction in HYGRO … SRIM case would be expected to reduce the AAOD", it is not clear to me why this is the case, please explain further.

6. (A general question and section 3.1.5) It is well known that relative humidity affects aerosol scattering through the size of the particles (aerosol swelling) and aerosol water content. These are related to the aerosol hygroscopicity which is determined by the aerosol mixing state. I would like to see some discussion about the role of relative humidity in the case of biomass burning (in this study), for example when the authors discuss aerosol-radiation interactions in 3.1.5 (p.16). For example, over the model domain, there is variation of relative humidity, how does the difference between 1L2B, HYGRO and SRIM relate to the relative humidity?

7. (Model infrastructure) It is expected to see in the methodology section about the construction of aerosol mixing state representation schemes of HYGRO and 1L2B. For example, how the aerosol particles are transferred from one bin to another. Please provide some descriptions in this aspect.

8. (Model computational aspect) It is expected to see some information of the computational cost and accuracy comparisons among the three aerosol mixing state representations. The accuracy could be calculated with respect to observations or SRIM.

---

## Referee Comment (RC2)

In this study, the authors implemented a three-category representation of aerosol mixing-state (1L2B) into an air quality and weather forecast model (GEM-MACH), which can better resolve differences in hygroscopicity due to aerosol composition and the amount of absorption enhancement of BC due to non-absorbing coatings. A two-category representation of aerosol mixing-state (HYGRO) was also implemented in the model. They found that the more-detailed representation of the aerosol hygroscopicity in both 1L2B and HYGRO decreases wet deposition, which increases aerosol concentrations, particularly of less-hygroscopic species. They further show that these increased aerosol concentrations increase cloud droplet number concentrations and cloud reflectivity in the model, decreasing surface temperatures. The manuscript is well written, and results are clearly presented. I have a few comments for the authors to consider.

General Comments:
1. It is a bit unclear to me how the aerosol species in each category are treated during physical and chemical processes. I would suggest the authors to make it clear and explain it a bit more in section 2. Some details can go to appendix or supplement.

I wonder how many aerosol species are in the three categories. Does each category all have 8 aerosol species? In each size bin, it has $8\times3=24$ aerosol species. In total, the model has $8\times3\times12=288$ aerosol species being transported/advected? Please give some more details.

The authors assume that aerosol in the lo-κ mixing-state categories does not participate in aqueous chemistry and is not removed by cloud-to-rain conversion and subsequent wet deposition. Can aerosols in the lo-κ mixing-state categories be activated as cloud droplet? Or the authors mean they can be activated but would not be allowed to be removed by in-cloud wet scavenging processes? How does the model treat the aerosols for droplet and rain evaporation processes? Does the model have some kind of "cloud borne" tracers to track these aerosol species?

I agree with reviewer #1 and would also suggest that the authors to explain in the manuscript how chemical reactions could also modify aerosol mixing state. According to section 2.1, emitted dust would go to lo-κ_lo-BC category. After gas-aerosol partitioning, sulfate, nitrate, and chloride can form on the particle surface making it more hygroscopic. All aerosol species in this category will be moved to hi-κ when reaching the threshold?

2. There are also uncertainties in the hygroscopicity values used for aerosol species. I would love to hear the authors' comments on the impact of the hygroscopicity uncertainties on the three-category aerosol mixing state treatment.

3. Regarding the analysis, I wonder if the authors have checked the vertical profiles of the

aerosol species and use some aircraft campaigns during the simulation period, like NASA ATom-1? The comparisons of aerosol species with IMPROVE and CSN tend to give similar results. To me, it would be more useful to compare concentrations of large aerosol particles as well (e.g., against EPA CASTNET measurements). Since the authors have looked at cloud properties like cloud droplet number concentrations, I wonder if the authors have further looked at the changes in the radiative fluxes or aerosol forcing.

Specific Comments:

For Figures 1-7, I don't see a strong need to organize the panels in this way. SRIM panel plots are much larger than the other panel plots, and there is much white space there. To me, three panel plots in a row looks more organized. There are no marks/lines showing latitudes and longitudes. Please add them. Although the authors stated that most variables shown are identical in the HYGRO and 1L2B simulations, it may still be useful to show the spatial distributions of differences/relative differences between 1L2B and HYGRO. Do 1L2B and HYGRO have identical spatial distributions of the variables shown.

Lines 43-58, consider adding a summary table for the various representations of aerosol mixing-state?

Lines 101-102, I'm quite confused here. Please try to rephrase/clarify it. What do you mean "our BC-free category is also hydrophobic"? If the authors mean hi-κ category, why it is also hydrophobic. Or the authors mean lo-κ_lo-BC category? "we have a single category for all hydrophilic particles" seems also to be hi-κ category?

Lines 142-143, "Therefore, there is never any absorption enhancement for particles in the hi-BC mixing-state category". The authors assume that there is no absorption enhancement for BC cores that comprise more than 40% of the particle by mass. The authors also stated previously that a threshold of 0.3 between lo-BC and hi-BC is used. I wonder what about the BC mass fraction between 0.3 and 0.4.

Lines 160-165, it is also not very clear to me. What do you mean "dust is defined to be the residual after the other components are accounted for"? Is the dust emission calculated online or prescribed?

---

## Referee Comment (RC3)

**Review of "An Improved Representation of Aerosol Mixing State for Air-Quality-Weather Interactions" by Stevens et al.**

**General comments:**

This work implements a detailed representation of aerosol mixing state, which is categorized by hygroscopicity and BC mass fraction, into the GEM-MACH air quality and weather forecast model and examines the impact of the aerosol mixing state representations on the aerosol concentrations, radiative properties, and meteorology over the North American region. The authors show that the more-detailed representation of the aerosol hygroscopicity (three-category 1L2B and two-category HYGRO) increases aerosol concentrations, which increases cloud droplet number concentrations and cloud reflectivity in the model. They also show that resolving thinner coatings on BC (1L2B) slightly decreases absorption aerosol optical depths.

This study is interesting and the subject is of great interest to ACP. The manuscript is well written. However, the model descriptions of the aerosol mixing state are not satisfactory, which are cause for concern (see Major comments). Once these points are addressed satisfactory, the paper should in my opinion be suitable for publication in ACP.

**Major comments:**

1. Section 2:

The authors should explain the treatment of aerosols in the mixing state categories in the model in more details. It is unclear to me. Do the model trace both mass and number concentrations of aerosols in each bin? How to calculate diameter of the aerosols in each bin (e.g., stationary or moving structures)? How to treat the movement of aerosols between different categories (e.g., moving center approach, all particles (mass of species and number) in the bin are moved to a new bin)? Do the condensations of sulphate onto the lo-κ_hi-BC and lo-κ_lo-BC categories move the aerosols in these categories to the hi-κ category? Can the aerosols in the low hygroscopicity categories (lo-κ_hi-BC and lo-κ_lo-BC) be act as CCN and be activated to cloud droplets? What schemes of the gas phase chemistry and the aerosol thermodynamics the model use? How to calculate dust emissions in the model? Could the authors explain the calculation methods of the aerosol optical properties of BC-containing particles in more detail (please also see specific comments)?

2. Abstract:

The authors describe that "the more-detailed representation of the aerosol hygroscopicity in both 1L2B and HYGRO decreases wet deposition, which increases aerosol concentrations, particularly of less-hygroscopic species" and "We show that these increased aerosol concentrations increase cloud droplet number concentrations and cloud reflectivity". I cannot understand that the increases in less-hygroscopic aerosol concentrations increase cloud droplet number concentrations. The less-hygroscopic species are not able to act as CCN and are not removed from the atmosphere by in-cloud scavenging process, but can be activated to cloud droplets in the model? Please clarify this.

3. Mixing state of BC-containing particles:

I wonder if the authors have validated the BC mass fractions of BC-containing particles in the model calculations by the available surface or aircraft observations. Some measurements, such as SP2 (single-particle soot photometer), would provide the information of BC mixing state, which can be used for the model validation (e.g., Oshima et al., 2009b). Although there are small amounts of data available on the BC mixing state, the authors should mention that they have validated the BC mixing state or not in their previous studies and, if not, please mention that the comparison should be done in future works.

4. Horizontal resolution:

The authors used the 10 km horizontal resolution in the model calculations. In my opinion, the 10 km resolution is high for simulations with the detailed aerosol mixing state. Can the authors state some benefits in their results due to the high resolutions?

**Specific comments:**

Lines 4 and 459, "This is the first model …" Is it the first model, because some models treat more detailed aerosol mixing state representations (e.g., Jacobson, 2001).

Lines 9-14, the results shown in this study will depend on the regions, season, and scenarios (e.g., anthropogenic and biomass burning). The authors should clarify these conditions in the abstract.

Introduction, it would be better to compare the results with Stier et al. (2006), which showed the decrease or enhancement of absorption due to BC aging, which increases the CCN activity and light absorption of BC-containing particles.

Line 101, "our BC-free category is also hydrophobic". I cannot understand this. The authors mean that lo-κ_lo-BC category is hydrophobic even if there is no BC in the lo-κ_lo-BC category? Does the hi-κ category include BC? Are there BC-free categories in 1L2B representation? Please clarify this.

Lines 111 and 114, "add the mass to the mixing-state category that matches the new particle's properties" and "the mass is moved to the mixing-state category that matches the hygroscopicity and BC mass fraction of the aerosol mass". How to treat number concentrations of aerosols?

Lines 137-138, "For the radiation calculations, seaspray and dust are always assumed to exist as pure particles, externally mixed from the other components". How to estimate number concentrations and diameters of seaspray and dust particles from the internally mixed aerosol representations?

Line 140, "The absorption enhancement of the BC cores is calculated following Bond et al. (2006)." Do the authors implement the Mie theory for concentrically coated spheres (core-shell treatment) and use complex refractive indices of individual chemical species? Or do the authors simply use the absorption enhancement factors for BC-containing particles in hi-κ and lo-κ_lo-BC categories? If in the latter case, how to calculate extinction (or scattering) coefficients of BC-containing particles and how to treat enhancement due to RH?

Line 147, "we perform simulations only from June 15th to July 31st, and." It is better to clarify the year (2016) here.

Lines 221 and 255, "all aerosol in the low-κ categories are excluded from wet deposition processes". Is this "in-cloud scavenging processes" rather than "wet deposition processes"? Coarse particles in the low-κ categories can removed from the atmosphere by below-cloud scavenging.

Figure 2 and lines 233-246, the authors show the temporal means of the wet deposition fluxes normalized by the surface aerosol concentrations. Do the authors show the wet deposition fluxes normalized by the aerosol column, if possible, because the in-cloud scavenging processes play efficiently at higher altitudes?

Lines 253 and 217, "After normalizing by the surface concentrations of aerosol, wet deposition rates of BC, POA, SOA, and dust were reduced in the HYGRO and 1L2B simulations by 27%, 40%, 12%, and 10%" and "These differences are due mostly to increases in less-hygroscopic species, with concentrations of BC, POA, SOA, and dust being increased in the HYGRO and 1L2B simulations by 16%, 16%, 21%, and 93%." Why the wet deposition rate of dust (10%) is smaller, although the surface concentrations of dust are largely increased?

Line 269, "We convert observed organic carbon to organic matter assuming a mass-to-carbon ratio of 1.8". Could the authors show references (ratio of 1.8) here?

Lines 315-322 and Figure 4, it is difficult to understand the BC transport and evaporation from Figure 4. Could the authors explain more clearly? How about the changes in BC column over the source (forest fires) and downwind regions?

Lines 374-379, why AAOD decrease in the Gulf of Mexico and just south the Great Lakes region? Are these regions close to the anthropogenic source regions? Is the resolving the coating thickness of BC important for freshly emitted anthropogenic BC particles? Could you add explanations?

It is helpful to show the emission map of BC (both anthropogenic and forest fires) during the model simulation period in Supplemental figures to clarify the source and remote regions.

**Technical comments:**

Table 3, I could not find the word "Table 3" in the text.

Lines 92-93, "0-2.5 um and 2.5-10 um". Are these diameters?

Line 101, "in that we our BC-free", remove "we", typo?

Line 310, "BC concentrations at the surface". BC mixing ratio?

Figure S1, please add units, like Figure 1.

**References:**

Jacobson, M. Z. (2001), Strong radiative heating due to the mixing state of black carbon in atmospheric aerosols, Nature, 409, 695–697.

Oshima, N., M. Koike, Y. Zhang, Y. Kondo, N. Moteki, N. Takegawa, and Y. Miyazaki (2009b), Aging of black carbon in outflow from anthropogenic sources using a mixing state resolved model: Model development and evaluation, J. Geophys. Res., 114, D06210, doi:10.1029/2008JD010680.

Stier, P., J. H. Seinfeld, S. Kinne, J. Feichter, and O. Boucher (2006), Impact of nonabsorbing anthropogenic aerosols on clear-sky atmospheric absorption, J. Geophys. Res., 111, D18201, doi:10.1029/2006JD007147.

---

## Author Comment (AC1)

We thank the referees for their comments on the manuscript. We respond to the referees' comments in a point-by-point fashion below. The referees' comments are bolded and italicised. New text added to the manuscript is italicised and written in blue font.

**Referee #2**

We note that Referee #2 provided their comments at an earlier review stage, and that we addressed their comments and edited the manuscript at that time. In the spirit of open peer review, we repeat our responses to their comments here, but we note that the discussion version of the manuscript has already been edited to address these comments. We have extended our response to Referee #2's sixth comment, based on further reflection.

*Summary*
*This study is about development of model representation of aerosol mixing state in chemical transport model GEM-MACH. The authors developed three-category representation (1L2B) to resolve aerosol population into three categories, high hygroscopicity, low hygroscopicity with high BC content, and low hygroscopicity with low BC content. The author further compared the 1L2B representation to two other representations called HYGRO (2-category, high hygroscopicity, low hygroscopicity) and SRIM (the original one in GEM-MACH which assume complete internally-mixed aerosols) by evaluating against the observation of a case of biomass-burning over North America. Besides, the study investigated the impact of aerosol mixing state representation on meteorology with and without feedback. The study matches the scope of ACP. I recommend acceptance of this manuscript for publication upon satisfactorily clarifications of the following issues.*

*Questions/clarifications*
*1. (Line 25) Apart from condensation and coagulation, chemical reactions could also modify aerosol mixing state. I suggest the authors to clarify this.*

We thank the reviewer for pointing this out. We have rephrased the sentence to the following:

*In general, particles emitted from different sources are initially externally mixed with respect to each other, and become more internally mixed with time through condensation, coagulation, and chemical reactions.*

*2. (Line 69-75) In the aerosol representation of 1L2B, the fixed bin boundary of hygroscopicity (kappa) and BC mass fraction of 0.1 and 0.3 proposed by Ching et al., 2016 was derived based on urban environments. It is possible that in the scenarios of biomass burning (test case in this study), such fixed boundary values may not be the most optimal in resolving aerosol mixing state. Should the author provide some sensitivities studies on the bin boundary positions, it would enhance the scientific values of this manuscript.*

We agree with the reviewer in principle that it would be useful to perform sensitivity studies where the mass-fraction and hygroscopicity boundaries are varied from the values chosen for this study, and this is something that we are interested in investigating in future work. However, given the resources necessary to perform our simulations, we feel that this question is beyond the scope of the current work.

Additionally, we note that BC mass fraction threshold used is similar to the BC mass fraction threshold of 0.4 at which our radiation scheme assumes that there is no absorption enhancement of BC (Majdzadeh et al., 2022), which was in turn chosen based on observations that show a pronounced change in the absorption enhancement of BC at around this mass ratio (Liu et al 2017, Peng et al 2016). We therefore believe that the optimal value of the BC mass fraction threshold should be similar to our chosen value of the BC mass fraction threshold.

**3. (Section 2.1) Apart from the atmospheric physical and chemical processes, the sensitivity study of aerosol mixing state also depends on mixing state of emitted aerosols. It is expected that the authors should state clearly or tabulate the assumed chemical composition of the particles from biomass burning (i.e. smoke particles from forest fire) as well as other aerosol types when they are emitted.**

We thank the reviewer for noticing that this important detail was missing from the text. The large number of source profiles prevents us from listing the chemical compositions of all aerosol source types to the text, but we have added information describing how the aerosol speciation was chosen, and we include the speciation profile for wildfire emissions as an example. For other specific sources, we direct the reviewer to Table S6 from Reff et al. (2009).
We have added the following to Sect 2.1:

*The Sparse Matrix Operator Kernel Emissions (SMOKE) emissions processing system (https://www.cmascenter.org/smoke; Bieser et al., 2001; Hogrefe et al., 2003; Houyoux et al., 2000) is used to speciate emissions prior to input within GEM-MACH (Zhang et al., 2018). Bulk aerosol mass emissions are associated with one of the 91 composite particulate matter speciation profiles compiled from the EPA's SPECIATE4.5 database (https://www.epa.gov/air-emissions-modeling/speciate-2; Reff et al., 2009). Each composite particulate matter speciation profile gives relative fractions of sulphate, nitrate, ammonium, black carbon, and POA. Sea salt and SOA are assumed to make no contribution, and dust is defined to be the residual after the other components are accounted for. As an example, particulate emissions from wildfires are speciated as 78.5\% POA, 9.7\% dust, 9.5\% BC, 1.3\% sulphate, 0.9\% ammonium, and 0.1\% nitrate.*

**4. (Line 303) According to the latest particle-resolved study published by Yao et al., 2022 ACP, based on 1800 environmental scenarios, simplified aerosol representation (assuming internally-mixed aerosol particles) could potentially to a wide range of underestimation in aerosol scattering with a maximum of -32% among the environment scenarios they studied. Therefore, I think it is better for the author to clarify in the text**

*that error in scattering coefficient of an aerosol population may not be negligible, instead depends on the interaction between the aerosol particles and the environment, such as relative humidity and chemistry processes involved.*

We thank the reviewer for noting that our original phrasing was too strong. However, we also note that for the 10 km spatial resolution monthly-mean values we show, a variety of aerosol conditions would be present for each grid column. The more appropriate comparison from Yao et al., (2022) would be the median change or the 25th percentile change in aerosol scattering, being -1% or -4%, respectively. We also note that the experiments of Yao et al., (2022) targeted populations of carbonaceous aerosol, excluding mineral dust and sea salt, and with median sulphate concentrations smaller than median BC concentrations. It is therefore possible that the relative effect on scattering in Yao et al., (2022) is larger than it would be for ambient aerosol populations in North America.

We have modified the sentence to indicate that we do not expect differences in our simulation, rather than to suggest that the representation of BC mass fraction cannot affect aerosol scattering in general.

We have added the following to the manuscript:

*Previous studies have found that the optical properties of non-absorbing aerosol is not strongly sensitive to the mixing-state of the aerosol (e.g. Zaveri et al., 2010; Klingmüller et al., 2014), and that because AOD is dominated by the scattering component, ambient AOD is not strongly sensitive to mixing-state (e.g. Matsui et al., 2013, 2014; Klingmüller et al., 2014; Han et al., 2013), although a recent study has shown that aerosol scattering can be very sensitive to aerosol mixing-state under certain conditions (Yao et al., 2022). We therefore do not expect our more-detailed representation of the BC mass to yield strong changes in aerosol scattering, but we do expect a decrease in aerosol absorption.*

**5. (Line 311) Please explain why "higher black carbon mass fraction in HYGRO ... SRIM case would be expected to reduce the AAOD", it is not clear to me why this is the case, please explain further.**

We thank the reviewer for showing us that we were unclear. If the mass of all aerosol species, including black carbon, was equal in both cases, an increase in the black carbon mass fraction in black carbon containing particles would imply that the black carbon had thinner coatings of non-absorbing material. These thinner coatings would be expected to result in a smaller absorption enhancement on the black carbon, and therefore a smaller AAOD. We have changed the text to the following to make our meaning clearer for the reader:

*If the mass concentrations of all aerosol species were equal in both cases, higher BC mass fractions would imply thinner coatings and smaller absorption enhancements for the BC-containing particles. This effect would be expected to reduce the AAOD in the HYGRO case*

*as compared to the SRIM case. The simulated increase in AAOD is due primarily to the increased concentrations of BC in the HYGRO case compared to the SRIM case.*

**6. (A general question and section 3.1.5) It is well known that relative humidity affects aerosol scattering through the size of the particles and aerosol water content. The later two factors are related to the aerosol hygroscopicity which is determined by the aerosol mixing state. I would like to see some discussion about the role of relative humidity in the case of biomass burning (in this study), for example when the authors discuss aerosol-radiation interactions in 3.1.5 (p.16). For example, over the model domain, there is variation of relative humidity, how does the difference between 1L2B, HYGRO and SRIM relate to the relative humidity?**

In order to address the reviewer's comment, we plot below two-dimensional histograms of the absolute and relative differences in the aerosol scattering optical depth (ASOD) for each grid column and 3-hourly mean (the time resolution of our saved data) against the surface specific humidity for the same grid column and 3-hourly mean. For completeness, we also show the same for the aerosol absorption optical depth (AAOD) below. We note that one of these is a column variable and the other is a surface variable, which means that the surface humidity will not be representative in cases where most of the aerosol is elevated above the surface, but we do not expect this to be the case generally. We print the correlation coefficient in between the two variables plotted in the upper right corner of each plot.

We note that the comparison between SRIM and the other two simulations is convoluted by the difference in aerosol mass between these simulations. We therefore restrain our discussion to the differences between the 1L2B and HYGRO simulations.

There appears to be a very weak negative correlation (R=-0.15) between the absolute and relative differences in ASOD and the surface specific humidity. We also note that most of the data points show small differences in ASOD: The 1st and 99th percentiles of the relative differences in ASOD are -0.6% and 1.4%, respectively. There is a stronger negative correlation (R=-0.47 or R=-0.32) between the AAOD and the surface specific humidity. This may be due water uptake on the particles enhancing the lensing effect.

We will respond to this comment further when we address the pending comments of the other two reviewers on the manuscript.

*We added the following when we responded to the other two referees:*

The radiation scheme used in this work, described in Majdzadeh et al. (2022), accounts for the effect of relative humidity on aerosol scattering (as well as absorption of non-BC species) through look-up tables of extinction efficiency $Q_{ext}$, single scattering albedo (ssa), and asymmetry parameter. Individual look-up tables were created for each species. The values of $Q_{ext}$, ssa, and the asymmetry parameter retrieved from these look-up tables for each species

depend on the relative humidity in the grid cell and the ratio between the dry aerosol radius and the wavelength.

Since the dependence of $Q_{ext}$ and ssa on relative humidity for each species does not depend on the quantity of any other species in the same particle, the interaction of the scattering efficiency $Q_{sca} = Q_{ext}$*ssa for each species with relative humidity will not depend on the amount of any other species in the same particle. Therefore, in the radiation scheme used in this work, a change in the relative humidity will have the same relative impact on aerosol scattering regardless of the mixing-state configuration.

[Figure]

[Figure]

**7. (Model infrastructure) It is expected to see in methodology about the construction of aerosol mixing state representation schemes of HYGRO and 1L2B. For example, how the aerosol particles are transferred from one bin to another. Please provide some descriptions in this aspect.**

We thank the reviewer for pointing out the omission of these details. We have added the following paragraph to Section 2:

*Coagulation of two particles within the same mixing-state category is assumed to result in a particle of the same mixing-state category, as both BC mass fraction and volume-weighted*

*hygroscopicity would be within the range spanned by the two original particles. For coagulation of particles from two different mixing-state categories, we calculate the hygroscopicity and BC mass fraction of the new particle, and add the mass to the mixing-state category that matches the new particle's properties. Additionally, after all other aerosol processes, we calculate the hygroscopicity and BC mass fraction for each size bin and mixing-state category. If either the hygroscopicity or the BC mass fraction is outside of the bounds of the current mixing-state category, the mass is moved to the mixing-state category that matches the hygroscopicity and BC mass fraction of the aerosol mass.*

**8. (Model computational aspect) It is expected to see some information of computational cost and accuracy comparisons among the three aerosol mixing state representations. Accuracy could be calculated with respect to observations or SRIM.**

We thank the reviewer for noting that we neglected to comment on computational cost in the text. We have added the following to our conclusions section:

*Currently, the HYGRO and 1L2B configurations require approximately 70% and 150% more running-time, respectively, than the SRIM configuration. We expect to reduce this additional cost through improvements to the efficiency of the model tracer transport scheme in the near future.*

We would like to draw the attention of the reviewer to Sect. 3.1.2 and Table 3, where we evaluate the accuracy of the SRIM and 1L2B simulations against observational data from IMPROVE, US EPA CSN and US EPA AQS. We also highlight this comparison in the third paragraph of our conclusions section. We note that the concentrations of aerosol species in the HYGRO and 1L2B simulations are very similar, and we do not expect that any evaluation of the accuracy of HYGRO would differ significantly from that of 1L2B.

**Referee #3**

*In this study, the authors implemented a three-category representation of aerosol mixing-state (1L2B) into an air quality and weather forecast model (GEM-MACH), which can better resolve differences in hygroscopicity due to aerosol composition and the amount of absorption enhancement of BC due to non-absorbing coatings. A two-category representation of aerosol mixing-state (HYGRO) was also implemented in the model. They found that the more-detailed representation of the aerosol hygroscopicity in both 1L2B and HYGRO decreases wet deposition, which increases aerosol concentrations, particularly of less-hygroscopic species. They further show that these increased aerosol concentrations increase cloud droplet number concentrations and cloud reflectivity in the model, decreasing surface temperatures. The manuscript is well written, and results are clearly presented. I have a few comments for the authors to consider.*

*General Comments:*
*1. It is a bit unclear to me how the aerosol species in each category are treated during physical and chemical processes. I would suggest the authors to make it clear and explain it a bit more in section 2. Some details can go to appendix or supplement.*

We explain this in more detail in our responses to the referee's other comments below.

*I wonder how many aerosol species are in the three categories.*
*Does each category all have 8 aerosol species? In each size bin, it has 8x3=24 aerosol species. In total, the model has 8x3x12=288 aerosol species being transported/advected? Please give some more details.*

The reviewer is correct; Each category has all eight aerosol species, and the 1L2B simulation has 288 aerosol tracers. We have added the following to Sect. 2 to clarify this:

*Each of the eight dry species are tracked in each mixing-state category, resulting in a total of 288 (8 species x 3 mixing-state categories x 12 size bins) aerosol tracers.*

*The authors assume that aerosol in the lo-k mixing-state categories does not participate in aqueous chemistry and is not removed by cloud-to-rain conversion and subsequent we deposition. Can aerosols in the lo-k mixing-state categories be activated as cloud droplet? Or the authors mean they can be activated but would not be allowed to be removed by in-cloud wet scavenging processes? How does the model treat the aerosols for droplet and rain evaporation processes? Does the model have some kind of "cloud borne" tracers to track these aerosol species?*

In order to make the treatment of the aerosol species more clear, we have added the following description to the supplement:

In GEM-MACH, the calculation of cloud droplet number concentrations ($N_C$) depends on whether or not aerosol feedbacks are enabled. In the simulations without aerosol-meteorology feedbacks, aerosol number concentrations have no effect on $N_C$ for the purposes of determining cloud-radiation interactions and all cloud microphysics processes, including rain formation. Instead, $N_C$ for meteorological processes is calculated using a single pre-specified, constant cloud condensation nuclei type and concentration at all points in time and space.

For the purposes of determining aerosol processes, including aqueous chemistry and transport or removal of aerosol within cloud droplets, a diagnostic $N_C$ is calculated based off of the total hydrophilic aerosol number concentration $N_A$, according to the parameterisation of Jones et al. (1994):

$$N_C = 375( 1 - exp( -2.5 \times 10^{-3} N_A ) )$$

The $N_C$ calculated in this way may differ from the $N_C$ used for meteorological processes. Here, $N_A$ is calculated by dividing the aerosol volume concentration in the hi-κ mixing-state category in each size bin by the volume of an aerosol particle with the midpoint diameter of the size bin and summing over the size bins. In the SRIM simulation, this reduces to the sum of the total aerosol volume concentration in each size bin divided by the the volume of an aerosol particle with the midpoint diameter of the size bin. The largest $N_C$ particles in the hi-κ mixing-state category are then selected to participate in in-cloud aerosol processes, including aqueous chemistry.

In the simulations with aerosol-meteorology feedbacks enabled, the algorithm described above differs in that $N_C$ is parameterized using Abdul-Razzak and Ghan (2002). Particle hygroscopicity is calculated separately for each mixing-state category based on molecular weights and ion dissociation, as per eq. 7 from Abdul-Razzak and Ghan (2002). Therefore, the aerosol mass in the lo-κ mixing-state categories is included when calculating $N_C$. The same value of $N_C$ is used both for meteorological processes and aerosol aqueous-phase processes.

However, the largest $N_C$ particles in the hi-κ mixing-state category are still selected to participate in in-cloud aerosol processes, neglecting the aerosol mass in the lo-κ mixing-state category(ies). This may lead to some unphysical behaviour where additional mass in the lo-κ mixing-state category(ies) can lead to smaller aerosol particles in the hi-κ mixing-state category participating in in-cloud aerosol processes, due to the increase in $N_C$. Investigating this behaviour was beyond the scope of the current work. We intend to improve on the representation of this process in a future version of GEM-MACH, so that the aerosol mass contributing to $N_C$ (in both hi-κ and lo-κ mixing-state categories) is consistent with the aerosol mass participating in in-cloud aerosol processes. This will allow for the possibility of large aerosol particles in lo-κ mixing-state categories to participate in in-cloud aerosol processing and wet deposition.

There are no cloud-borne aerosol tracers transported between chemistry timesteps. Instead, aerosol mass is activated as described above. For this portion of the aerosol mass only, aqueous chemistry is calculated, and cloud-to-rain conversion followed by either downwards transport by evaporating precipitation or wet deposition to the surface is accounted for. Afterwards, the new in-cloud aerosol mass is transferred back to the aerosol tracers.

*I agree with reviewer #1 and would also suggest that the authors to explain in the manuscript how chemical reactions could also modify aerosol mixing state. According to section 2.1, emitted dust would go to lo-k_lo-BC category. After gas-aerosol partitioning, sulfate, nitrate, and chloride can form on the particle surface making it more hygroscopic. All aerosol species in this category will be moved to hi-k when reaching the threshold?*

The reviewer is correct: aerosol-gas partitioning and heterogeneous chemistry allow the formation of sulphate, nitrate, and ammonia on dust particles (we do not currently include chloride partitioning), making the dust particles more hygroscopic. If the volume-weighted hygroscopicity increases beyond the threshold, all mass in that combination of mixing-state category and size bin will be moved to the hi-κ mixing-state category, including the dust mass. However, we note that the heterogeneous chemistry mechanism currently implemented in GEM-MACH is a bulk scheme based on ISOROPPIA (Makar et al., 2003) that does not depend in any way on the mass of dust. The gas-particle partitioning rates of sulphate, nitrate, and ammonium do not depend on their distribution among size bins or mixing-state categories.

We have added references for the chemistry mechanisms use in GEM-MACH to Sect. 2, and we now note that the heterogeneous chemistry is a bulk scheme. We have also updated our description of the transfer of mass between mixing-state categories to the following to make this more clear:

*Coagulation of two particles within the same mixing-state category is assumed to result in a particle of the same mixing-state category, as both BC mass fraction and volume-weighted hygroscopicity would be within the range spanned by the two original particles. For coagulation of particles from two different mixing-state categories, we calculate the hygroscopicity and BC mass fraction of the new particle, and add the mass to the mixing-state category that matches the new particle's properties. No other process directly transfers mass between mixing-state categories. However, after all other aerosol processes, we calculate the hygroscopicity and BC mass fraction for each size bin and mixing-state category. If either the hygroscopicity or the BC mass fraction is outside of the bounds of the current mixing-state category, all of the mass in the current combination of size bin and mixing-state category is moved to the mixing-state category that matches the hygroscopicity and BC mass fraction of the aerosol mass. Through this method, as condensation and other processes change the volume-weighted hygroscopicity and BC mass fraction over time, aerosol particles will generally move from the lo-κ_hi-BC mixing-state category to the lo-κ_lo-BC mixing-state category, and from both lo-κ mixing-state categories to the hi-κ mixing-state category.*

*2. There are also uncertainties in the hygroscopicity values used for aerosol species. I would love to hear the authors' comments on the impact of the hygroscopicity uncertainties on the three-category aerosol mixing state treatment.*

There are two locations in the model where aerosol hygroscopicity is used: First, in the partitioning of aerosol between the hi-κ and lo-κ mixing-state categories. Second, only in the simulations with aerosol-meteorology feedbacks, in the calculation of $N_C$ as described above.

Regarding the partitioning of aerosol species between categories: Assuming that the threshold hygroscopicity between the hi-κ and lo-κ mixing-state categories is held fixed, an underestimate in the hygroscopicity of the less-hygroscopic species (BC, POA, and dust) would lead to too much of the mass of these species remaining in the lo-κ mixing-state categories. Under our current implementation, this would mean an underestimate in the wet deposition of these species, an overestimate in the atmospheric concentrations of these species, and all differences in results between the SRIM simulation and either the HYGRO or 1L2B simulations would be overstated. An overestimate in the hygroscopicities of these species would have the opposite effect: overestimated partitioning to the hi-κ category, overestimated wet deposition and atmospheric concentrations, and understated differences between the results of SRIM and either HYGRO or 1L2B.

If the hygroscopicities of each species were altered independently, we would expect that the hygroscopicity of dust would only have small effects on BC and POA concentrations. Most dust mass exists in larger size bins than most BC and POA mass, so we would expect that e.g. more-hydrophilic dust would have a small effect on the partitioning of BC and POA between mixing-state categories. We would also not expect the hygroscopicity of BC to strongly alter partitioning of POA, as concentrations of POA are much larger than BC, and most of the POA mass will exist in the lo-κ, lo-BC category rather than the lo-κ, hi-BC category. However, due to the co-emission of BC and POA, and the larger concentrations of POA than BC, we do expect that partitioning of BC could be sensitive to the assumed hygroscopicity of POA. A larger assumed hygroscopicity of POA could increase the transfer of BC to the hi-κ category and subsequent wet-deposition of BC.

One additional limiting case is worth considering as a thought experiment: if the hygroscopicities of POA and dust were similar to those of sulphate, nitrate, and ammonia, then all mass in the lo-κ, lo-BC category would transfer to the hi-κ category. The 1L2B mixing-state configuration would then reduce to a configuration with one category for lo-κ, thinly-coated BC and another category for all other aerosol. In this scenario, BC could not become thickly-coated without also becoming hydrophilic, so the two effects would become conflated.

Regarding the calculation of $N_C$: We will start this discussion by noting that the review by Stevens and Dastoor (2019) summarised previous studies examining the effects of mixing-state representation on cloud condensation nuclei (CCN) concentrations as "a size-resolved internally-mixed representation has been shown to frequently overestimate CCN concentrations by 10% to 20%". However, we note that in our analysis, we found that the 1L2B simulation had $N_C$ values ~15% higher than the SRIM simulation, due to larger masses of aerosol available to activate, in turn due to reduced wet deposition. An underestimation or overestimation in the hygroscopicity of one aerosol species (or all aerosol species) would lead to an underestimate or overestimate of $N_C$, regardless of the representation of the aerosol mixing-state. The difference

between the SRIM and 1L2B simulations would be most affected by the diversity in κ values among species: if the κ values for BC, POA, and dust were larger than we assumed (more similar to sulphate, nitrate, ammonia, sea-spray, and SOA) only for the calculation of $N_C$, then the difference in $N_C$ between SRIM and 1L2B would be greater, as $N_C$ would increase more in 1L2B than in SRIM. However, the effects of hygroscopicity errors on aerosol mass described above would have the opposite effect on $N_C$, and we expect that they would dominate the total effect on $N_C$.

*3. Regarding the analysis, I wonder if the authors have checked the vertical profiles of the aerosol species and use some aircraft campaigns during the simulation period, like NASA ATom-1? The comparisons of aerosol species with IMPROVE and CSN tend to give similar results. To me, it would be more useful to compare concentrations of large aerosol particles as well (e.g., against EPA CASTNET measurements). Since the authors have looked at cloud properties like cloud droplet number concentrations, I wonder if the authors have further looked at the changes in the radiative fluxes or aerosol forcing.*

We thank the reviewer for these excellent suggestions. We intend to focus on a more detailed evaluation in a future study, including an evaluation against observations of AOD. We intended the present study to focus on a description of the design and behaviour the mixing-state resolved configurations of GEM-MACH, especially with comparison to the original size-resolved internally-mixed version of GEM-MACH. For the present, we can speculate that if we compared against the EPA CASTNET measurements, the HYGRO and 1L2B configurations would perform more poorly in comparison to SRIM, due to the currently unrealistically low wet deposition rates of large dust aerosol. However, this will be improved by implementing the method of calculating dust activation into cloud droplets using Abdul-Razzak and Ghan (2002) and applying this for in-cloud aerosol processing in a future version of GEM-MACH.

*Specific Comments:*
*For Figures 1-7, I don't see a strong need to organize the panels in this way. SRIM panel plots are much larger than the other panel plots, and there is much white space there. To me, three panel plots in a row looks more organized. There are no marks/lines showing latitudes and longitudes. Please add them. Although the authors stated that most variables shown are identical in the HYGRO and 1L2B simulations, it may still be useful to show the spatial distributions of differences/relative differences between 1L2B and HYGRO. Do 1L2B and HYGRO have identical spatial distributions of the variables shown.*

We have reorganised the plots and added longitude and latitude lines as suggested. In addition, we have rearranged the order of the subplots in Figures 7 and 10 so that they are consistent with the new versions of the other figures (differences between the same two cases in a single row, absolute differences in the second column, relative differences in the third column).

We show below plots of the absolute and relative differences between 1L2B and HYGRO for $PM_{2.5}$, $AQHI_{2.5}$, BC, AOD, $PM_{10}$, and $AQHI_{10}$. Note that we use finer-scale colour bars for these plots than those comparing SRIM and HYGRO: Absolute difference colour bar ranges are

frequently one or two orders of magnitude smaller, and relative difference colour bar ranges are a factor of 10 smaller. Local relative differences in all variables are generally within +/- 2% except for the normalized wet deposition. The relative differences in normalized wet deposition are greatest over Labrador, but we note that this is region with low aerosol concentrations. Due to the normalization of dividing the wet deposition rates by the aerosol concentrations, small differences in the aerosol concentrations will yield large differences in the normalized wet deposition rates. Relative differences in other variables tend to be greatest over the Atlantic ocean, with the exception of AOD which is greatest over the eastern USA.

We have added these figures to the supplement, except for the differences in $PM_{10}$ and $AQHI_{10}$. For these two variables, we have instead updated the previous figures in the supplement to include the 1L2B-HYGRO differences.

[Figure]

**1L2B - hygro**

[Figure]

[Figure]

PM$_{2.5}$ AQHI absolute difference

[Figure]

PM$_{2.5}$ AQHI relative difference [%]

**1L2B - hygro**

[Figure]

BC absolute difference [$\mu$g kg$^{-1}$]

[Figure]

BC relative difference [%]

[Figure]

[Figure]

[Figure]

*Lines 43-58, consider adding a summary table for the various representations of aerosol mixing-state?*

We thank the reviewer for the suggestion. We have added the requested table.

*Lines 101-102, I'm quite confused here. Please try to rephrase/clarify it. What do you mean "our BC-free category is also hydrophobic"? If the authors mean hi-k category, why it is also hydrophobic. Or the authors mean lo-k_lo-BC category? "we have a single category for all hydrophilic particles" seems also to be hi-k category?*

We thank the reviewer for showing us that the text was unclear. When we say that our BC-free category is also hydrophobic, we are referring to the lo-κ_lo-BC category. We have rephrased the text to the following to make our meaning more clear:

*We differ in that we use a single category for all hydrophilic particles (hi-κ) and we use two categories for hydrophobic particles (lo-κ_hi-BC and lo-κ_lo-BC). This allows us to resolve BC coated with organic material (weakly hygroscopic, but thickly-coated) from BC that has thin coatings or no coatings of other aerosol matter (weakly hygroscopic and thinly-coated).*

*Lines 142-143, "Therefore, there is never any absorption enhancement for particles in the hi-BC mixing-state category". The authors assume that there is no absorption enhancement for BC cores that comprise more than 40% of the particle by mass. The authors also stated previously that a threshold of 0.3 between lo-BC and hi-BC is used. I wonder what about the BC mass fraction between 0.3 and 0.4.*

The statement was in error, and has been removed. The referee is correct, there would be absorption enhancement for particles in the lo-κ_hi-BC with a BC mass fraction between 0.3 and 0.4. We thank the referee for noticing this error and bringing it to our attention.

**Lines 160-165, it is also not very clear to me. What do you mean "dust is defined to be the residual after the other components are accounted for"? Is the dust emission calculated online or prescribed?**

We thank the referee for showing us that we were unclear. Anthropogenic dust emissions are sourced as other anthropogenic emissions. The SPECIATE4.5 database contains 37 trace element species, in addition to sulphate, nitrate, ammonium, BC, and organic aerosol species. After we account for the other components (sulphate, nitrate, ammonium, BC, and POA), we aggregate the contribution from all other species listed in the SPECIATE4.5 database and include them in our dust category. GEM-MACH does not currently include natural dust emissions; the only natural dust included in our simulations is that included in the boundary conditions. We have rephrased the text to the following to make our meaning more clear:

*Each composite particulate matter speciation profile gives relative fractions of sulphate, nitrate, ammonium, BC, and POA. Dust is defined to be the sum of all remaining species in the profile after these components are accounted for. Sea salt and SOA are assumed to make no contribution. [...] GEM-MACH does not currently include natural dust emissions; the only natural dust included in our simulations is that included in the boundary conditions.*

**Referee #4**

*General comments:*
*This work implements a detailed representation of aerosol mixing state, which is categorized by hygroscopicity and BC mass fraction, into the GEM-MACH air quality and weather forecast model and examines the impact of the aerosol mixing state representations on the aerosol concentrations, radiative properties, and meteorology over the North American region. The authors show that the more-detailed representation of the aerosol hygroscopicity (three-category 1L2B and two-category HYGRO) increases aerosol concentrations, which increases cloud droplet number concentrations and cloud reflectivity in the model. They also show that resolving thinner coatings on BC (1L2B) slightly decreases absorption aerosol optical depths.*

*This study is interesting and the subject is of great interest to ACP. The manuscript is well written. However, the model descriptions of the aerosol mixing state are not satisfactory, which are cause for concern (see Major comments). Once these points are addressed satisfactory, the paper should in my opinion be suitable for publication in ACP.*

*Major comments:*
*1. Section 2:*
*The authors should explain the treatment of aerosols in the mixing state categories in the model in more details. It is unclear to me. Do the model trace both mass and number concentrations of aerosols in each bin? How to calculate diameter of the aerosols in each bin (e.g., stationary or moving structures)?*

We thank the reviewer for showing us that we have not included sufficient detail for a clear understanding of the GEM-MACH model. GEM-MACH does not have a prognostic aerosol number tracer. Instead, aerosol number concentrations in each size bin are calculated from the aerosol mass by first dividing by the density of each species to obtain the aerosol volume concentration, and then dividing the aerosol volume concentration by the volume of a single particle with radius equal to midpoint radius of the size bin. We have added the following sentence to Sect. 2 to make this more clear, and the calculation of aerosol number concentrations is now described in the supplement (see below).

*GEM-MACH uses a single-moment aerosol scheme; It does not include a prognostic aerosol number tracer. When needed, diagnostic number concentrations are calculated assuming that aerosol particles within each size bin are monodisperse with diameter equal to the midpoint diameter of the size bin.*

*How to treat the movement of aerosols between different categories (e.g., moving center approach, all particles (mass of species and number) in the bin are moved to a new bin)?*

***Do the condensations of sulphate onto the lo-κ_hi-BC and lo-κ_lo-BC categories move the aerosols in these categories to the hi-κ category?***

For each size bin, the mass of sulphate added to each mixing-state category through condensation is distributed proportionally to the amount of mass in that mixing-state category. Then movement of aerosol between size bins due to condensation is treated using a moving bin approach, as described in eq. 3a and 3b of Gong et al. (2006), applied independently for each mixing-state category. This does not change the mixing-state category of the mass moved to the new size bin; the mass is distributed between the larger size bin and the current size bin, both in the same mixing-state category. After all other aerosol processes, we calculate the hygroscopicity and BC mass fraction for each combination of size bin and mixing-state category. If either the hygroscopicity or the BC mass fraction is outside of the bounds of the current mixing-state category, all mass in that combination of size bin and mixing-state category is moved to the mixing-state category that matches the hygroscopicity and BC mass fraction of the aerosol mass. Therefore, all processes that change the volume-weighted hygroscopicity (including condensation of sulphate) can result in the aerosol being moved to a different mixing-state category. In the case of sulphate condensation, this will increase the hygroscopicity of aerosol in the lo-κ mixing-state categories, and a sufficient mass of sulphate will cause the mass to be moved to the hi-κ mixing-state category. We have altered the text added in response to Referee #2 in order to attempt to make this more clear.

*Coagulation of two particles within the same mixing-state category is assumed to result in a particle of the same mixing-state category, as both BC mass fraction and volume-weighted hygroscopicity would be within the range spanned by the two original particles. For coagulation of particles from two different mixing-state categories, we calculate the hygroscopicity and BC mass fraction of the new particle, and add the mass to the mixing-state category that matches the new particle's properties. No other process directly transfers mass between mixing-state categories. However, after all other aerosol processes, we calculate the hygroscopicity and BC mass fraction for each size bin and mixing-state category. If either the hygroscopicity or the BC mass fraction is outside of the bounds of the current mixing-state category, all of the mass in the current combination of size bin and mixing-state category is moved to the mixing-state category that matches the hygroscopicity and BC mass fraction of the aerosol mass. Through this method, as condensation and other processes change the volume-weighted hygroscopicity and BC mass fraction over time, aerosol particles will generally move from the lo-κ_hi-BC mixing-state category to the lo-κ_lo-BC mixing-state category, and from both lo-κ mixing-state categories to the hi-κ mixing-state category.*

***Can the aerosols in the low hygroscopicity categories (lo-κ_hi-BC and lo-κ_lo-BC) be act as CCN and be activated to cloud droplets?***

We thank the reviewer for pointing out that this is a point of confusion. We have added the following description to the supplement to make the treatment of aerosol-cloud interactions more clear:

*In GEM-MACH, the calculation of cloud droplet number concentrations ($N_C$) depends on whether or not aerosol feedbacks are enabled. In the simulations without aerosol-meteorology feedbacks, aerosol number concentrations have no effect on $N_C$ for the purposes of determining cloud-radiation interactions and all cloud microphysics processes, including rain formation. Instead, $N_C$ for meteorological processes is calculated using a single pre-specified, constant cloud condensation nuclei type and concentration at all points in time and space.*

*For the purposes of determining aerosol processes, including aqueous chemistry and transport or removal of aerosol within cloud droplets, a diagnostic $N_C$ is calculated based off of the total hydrophilic aerosol number concentration $N_A$, according to the parameterisation of Jones et al. (1994):*

$$N_C = 375( 1 - exp( -2.5 \times 10^{-3} N_A ) )$$

*The $N_C$ calculated in this way may differ from the $N_C$ used for meteorological processes. Here, $N_A$ is calculated by dividing the aerosol volume concentration in the hi-κ mixing-state category in each size bin by the volume of an aerosol particle with the midpoint diameter of the size bin and summing over the size bins. In the SRIM simulation, this reduces to the sum of the total aerosol volume concentration in each size bin divided by the the volume of an aerosol particle with the midpoint diameter of the size bin. The largest $N_C$ particles in the hi-κ mixing-state category are then selected to participate in in-cloud aerosol processes, including aqueous chemistry.*

*In the simulations with aerosol-meteorology feedbacks enabled, the algorithm described above differs in that $N_C$ is parameterized using Abdul-Razzak and Ghan (2002). Particle hygroscopicity is calculated separately for each mixing-state category based on molecular weights and ion dissociation, as per eq. 7 from Abdul-Razzak and Ghan (2002). Therefore, the aerosol mass in the lo-κ mixing-state categories is included when calculating $N_C$. The same value of $N_C$ is used both for meteorological processes and aerosol aqueous-phase processes.*

*However, the largest $N_C$ particles in the hi-κ mixing-state category are still selected to participate in in-cloud aerosol processes, neglecting the aerosol mass in the lo-κ mixing-state category(ies). This may lead to some unphysical behaviour where additional mass in the lo-κ mixing-state category(ies) can lead to smaller aerosol particles in the hi-κ mixing-state category participating in in-cloud aerosol processes, due to the increase in $N_C$. Investigating this behaviour was beyond the scope of the current work. We intend to improve on the representation of this process in a future version of GEM-MACH, so that the aerosol mass contributing to $N_C$ (in both hi-κ and lo-κ mixing-state categories) is consistent with the aerosol mass participating in in-cloud aerosol processes. This will allow for the possibility of large aerosol particles in lo-κ mixing-state categories to participate in in-cloud aerosol processing and wet deposition.*

**What schemes of the gas phase chemistry and the aerosol thermodynamics the model use?**

We have added the following to Sect. 2 to answer this question:

*The gas-phase and aqueous-phase chemistry mechanisms in GEM-MACH are adapted from ADOM (Acid Deposition and Oxidant Model, Venkatram et al., 1988; Fung et al., 1991). The*

*heterogeneous chemistry mechanism currently implemented in GEM-MACH is a bulk scheme based on ISOROPPIA (Makar et al. 2003).*

**How to calculate dust emissions in the model?**

We thank the referee for showing us that we were unclear. Anthropogenic dust emissions are sourced as other anthropogenic emissions. The SPECIATE4.5 database contains 37 trace element species, in addition to sulphate, nitrate, ammonium, BC, and organic aerosol species. After we account for the other components (sulphate, nitrate, ammonium, BC, and POA), we aggregate the contribution from all other species listed in the SPECIATE4.5 database and include them in our dust category. GEM-MACH does not currently include natural dust emissions; the only natural dust included in our simulations is that included in the boundary conditions. We have rephrased the text to the following to make our meaning more clear:

*Each composite particulate matter speciation profile gives relative fractions of sulphate, nitrate, ammonium, BC, and POA. Dust is defined to be the sum of all remaining species in the profile after these components are accounted for. Sea salt and SOA are assumed to make no contribution. [...] GEM-MACH does not currently include natural dust emissions; the only natural dust included in our simulations is that included in the boundary conditions.*

**Could the authors explain the calculation methods of the aerosol optical properties of BC-containing particles in more detail (please also see specific comments)?**

We thank the referee for noting that this was unclear. We address the specific comments individually below. We hope that our responses to the specific comments address the reviewer's concerns. We also note that a full description of the radiation scheme is available in Majdzadeh et al., (2022).

**2. Abstract:**
**The authors describe that "the more-detailed representation of the aerosol hygroscopicity in both 1L2B and HYGRO decreases wet deposition, which increases aerosol concentrations, particularly of less-hygroscopic species" and "We show that these increased aerosol concentrations increase cloud droplet number concentrations and cloud reflectivity". I cannot understand that the increases in less-hygroscopic aerosol concentrations increase cloud droplet number concentrations. The less-hygroscopic species are not able to act as CCN and are not removed from the atmosphere by in-cloud scavenging process, but can be activated to cloud droplets in the model? Please clarify this.**

We thank the referee for showing us that we were not clear. We believe that the additional text added to the supplement, printed above in response to one of the referee's previous questions, will clarify this point.

**3. Mixing state of BC-containing particles:**

*I wonder if the authors have validated the BC mass fractions of BC-containing particles in the model calculations by the available surface or aircraft observations. Some measurements, such as SP2 (single-particle soot photometer), would provide the information of BC mixing state, which can be used for the model validation (e.g., Oshima et al., 2009b). Although there are small amounts of data available on the BC mixing state, the authors should mention that they have validated the BC mixing state or not in their previous studies and, if not, please mention that the comparison should be done in future works.*

We thank the reviewer for these excellent suggestions. As this is the first description of GEM-MACH modified to resolve mixing-state, no previous validation of e.g. BC mass fractions has been done. We intend to focus on a more detailed evaluation in a future study.

*4. Horizontal resolution:*
*The authors used the 10 km horizontal resolution in the model calculations. In my opinion, the 10 km resolution is high for simulations with the detailed aerosol mixing state. Can the authors state some benefits in their results due to the high resolutions?*

We thank the reviewer for giving us this opportunity to express the benefits of high horizontal resolution. We note that the high resolution is beneficial for distinguishing regions of fresh emissions, such as thinly-coated BC, from those where particles have aged and become coated with other aerosol matter. We have added the following text to Sect. 3.1.4, where we discuss the BC mass fraction in BC-containing particles:

*The high spatial resolution of GEM-MACH is an asset in resolving these regions close to emission sources.*

*Specific comments:*
*Lines 4 and 459, "This is the first model ..." Is it the first model, because some models treat more detailed aerosol mixing state representations (e.g., Jacobson, 2001).*

We appreciate that previous studies have used mixing-state representations of similar or greater complexity to the one we have chosen, and we certainly don't want to diminish the importance of their work. We have therefore removed this sentence from our abstract and our conclusions. We do feel that what we have implemented here is unique for a continental-scale model. Our representation is most similar to that described by Ching et al. (2016), to whom we attribute our inspiration, but that study did not use a continental-scale model. Our representation is also similar to the MADE-soot (Riemer et al., 2003; Vogel et al., 2009), MADE-in (Aquila et al., 2011) and MADE-3 (Kaiser et al., 2019, 2014) aerosol modules; we describe the differences between our approach and theirs in detail in Sect. 2.

*Lines 9-14, the results shown in this study will depend on the regions, season, and scenarios (e.g., anthropogenic and biomass burning). The authors should clarify these conditions in the abstract.*

We agree with the referee that our results are sensitive to our chosen domain and case study. We have added the following sentence to the abstract to describe our case study:

*We perform a case study focused on North America during July 2016, when there were intense wildfires over northwestern North America.*

**Introduction, it would be better to compare the results with Stier et al. (2006), which showed the decrease or enhancement of absorption due to BC aging, which increases the CCN activity and light absorption of BC-containing particles.**

As the referee did not give a line number for this comment, we are not certain which statement in our introduction they wish us to compare to the results of Stier et al. (2006). We note that Stier et al. (2006) did not report either CCN concentrations or absorption enhancement factors, though they did report effects of anthropogenic sulphate emissions on BC concentrations and AAOD.

**Line 101, "our BC-free category is also hydrophobic". I cannot understand this. The authors mean that lo-κ_lo-BC category is hydrophobic even if there is no BC in the lo-κ_lo-BC category? Does the hi-κ category include BC? Are there BC-free categories in 1L2B representation? Please clarify this.**

We thank the reviewer for showing us that the text was unclear. When we say that our BC-free category is also hydrophobic, we are referring to the lo-κ_lo-BC category. We have rephrased the text to the following to make our meaning more clear:

*We differ in that we use a single category for all hydrophilic particles (hi-κ) and we use two categories for hydrophobic particles (lo-κ_hi-BC and lo-κ_lo-BC). This allows us to resolve BC coated with organic material (weakly hygroscopic, but thickly-coated) from BC that has thin coatings or no coatings of other aerosol matter (weakly hygroscopic and thinly-coated).*

**Lines 111 and 114, "add the mass to the mixing-state category that matches the new particle's properties" and "the mass is moved to the mixing-state category that matches the hygroscopicity and BC mass fraction of the aerosol mass". How to treat number concentrations of aerosols?**

We thank the reviewer for showing us that we have not included sufficient detail for a clear understanding of the GEM-MACH model. GEM-MACH does not have a prognostic aerosol number tracer. Instead, aerosol number concentrations in each size bin are calculated from the aerosol mass by first dividing by the density of each species to obtain the aerosol volume concentration, and then dividing the aerosol volume concentration by the volume of a single particle with radius equal to midpoint radius of the size bin. We have added the following sentence to Sect. 2 to make this more clear, and the calculation of aerosol number concentrations is now described in the supplement (see above).

*GEM-MACH uses a single-moment aerosol scheme; It does not include a prognostic aerosol number tracer. When needed, diagnostic number concentrations are calculated assuming that aerosol particles within each size bin are monodisperse with diameter equal to the midpoint diameter of the size bin.*

**Lines 137-138, "For the radiation calculations, seaspray and dust are always assumed to exist as pure particles, externally mixed from the other components". How to estimate number concentrations and diameters of seaspray and dust particles from the internally mixed aerosol representations?**

Similar to the calculation of number concentrations for the purposes cloud-aerosol interactions, the dry diameters of sea-spray and dust are assumed to be the midpoint diameters of the size bins. For the radiation calculations, the number concentrations of seaspray and dust are calculated by dividing the masses of seaspray or dust in each size bin by the volume of a particle with the midpoint diameter of the size bin multiplied by a fixed density of 2.170 g cm$^{-3}$ for seaspray or 2.650 g cm$^{-3}$ for dust, respectively.

**Line 140, "The absorption enhancement of the BC cores is calculated following Bond et al. (2006)." Do the authors implement the Mie theory for concentrically coated spheres (core-shell treatment) and use complex refractive indices of individual chemical species? Or do the authors simply use the absorption enhancement factors for BC-containing particles in hi-κ and lo-κ_lo-BC categories? If in the latter case, how to calculate extinction (or scattering) coefficients of BC-containing particles and how to treat enhancement due to RH?**

We thank the referee for noting that this was unclear. We have altered the description of the aerosol-radiation interactions to the following text:

*Aerosol-radiation interactions are calculated as described in Majdzadeh et al. (2022). For the radiation calculations, sea-spray and dust are always assumed to exist as pure particles, externally mixed from the other components. For each size bin within each mixing-state category, if the BC mass fraction is greater than 40 %, BC is also assumed to be externally mixed from other components. For each size bin within each mixing-state category, sulphate, nitrate, ammonium and organic matter are assumed to be internally mixed. If the BC mass fraction is less than 40 % we assume that these same species (sulphate, nitrate, ammonium, POA and SOA) form a spherical shell over a spherical BC core in order to calculate an absorption enhancement factor. The total wet radius of the core-shell particle is calculated using the volume-weighted hygroscopic growth factor of the components in the core-shell particle. The absorption enhancement of the BC cores is then calculated following the parameterization of Bond et al. (2006) with the observationally constrained maximum threshold of 1.93 (Bond et al., 2006). However, we assume that there is no absorption enhancement for BC cores that comprise more than 40 % of the particle by mass, in agreement with more recent observations*

*of thinly-coated BC particles (Liu et al., 2017; Peng et al., 2016). This process is applied independently to each size bin and each mixing-state category.*

**Line 147, "we perform simulations only from June 15th to July 31st, and." It is better to clarify the year (2016) here.**

We have added the year as requested.

**Lines 221 and 255, "all aerosol in the low-κ categories are excluded from wet deposition processes". Is this "in-cloud scavenging processes" rather than "wet deposition processes"? Coarse particles in the low-κ categories can removed from the atmosphere by below-cloud scavenging**

We thank the referee for noting that we were imprecise with our language here. We have changed the phrasing here, and in other relevant parts of the manuscript, from "wet deposition" to "in-cloud scavenging".

**Figure 2 and lines 233-246, the authors show the temporal means of the wet deposition fluxes normalized by the surface aerosol concentrations. Do the authors show the wet deposition fluxes normalized by the aerosol column, if possible, because the in-cloud scavenging processes play efficiently at higher altitudes?**

Unfortunately, due to the large size of our output files, we did not save model output of the total aerosol column. We did recalculate the wet deposition fluxes normalized by the available column sum of the mixing ratios of each species (at the surface and at model hybrid levels between 0.807 and 0.962, approximately 35-185 hPa below surface pressure). We did not weight the mixing ratios by the layer thicknesses, we simply summed them for this calculation. We show the calculated normalized wet depositions fluxes below. Note the finer resolution of the colour scales on the 1L2B-HYGRO subplots compared to the HYGRO-SRIM subplots.

As shown, the wet deposition fluxes normalized by the available column sum are more spatially uniform than those of the wet deposition fluxes normalized by the surface concentrations. Notably, the anomalously large values in the northern reaches of the domain have been reduced to values more similar to the rest of the domain.

The differences between cases in this metric show similar patterns to those shown previously. The increases between HYGRO and SRIM in the southern part of the domain remain, but are reduced in extent. Perhaps clouds in this region are still at higher altitudes than our available aerosol data. The decreases in the northern part of the domain have reduced in magnitude.

[Figure]

**Lines 253 and 217, "After normalizing by the surface concentrations of aerosol, wet deposition rates of BC, POA, SOA, and dust were reduced in the HYGRO and 1L2B simulations by 27%, 40%, 12%, and 10%" and "These differences are due mostly to increases in less-hygroscopic species, with concentrations of BC, POA, SOA, and dust being increased in the HYGRO and 1L2B simulations by 16%, 16%, 21%, and 93%."**
**Why the wet deposition rate of dust (10%) is smaller, although the surface concentrations of dust are largely increased?**

We interpret the referee's question as being why the relative increase in dust concentrations are larger than those of BC, POA, or SOA, when the relative decrease in normalized wet deposition was smaller than for the same species. As the referee points out in a previous comment, wet deposition rates will depend on the vertical profile of the aerosol concentrations. If the dust vertical profile differed between the SRIM and HYGRO cases, with relatively more dust at the surface in the HYGRO case, this would lead to the changes in our normalized wet deposition rates being understated, compared to wet deposition normalized by in-cloud aerosol mass. This may be likely as a result of the decrease in wet deposition.

We show in the following figures the available mean vertical profiles for POA, SOA, BC, and dust. The mass mixing ratios of POA, SOA, and BC all increase sharply towards the surface, while the mass mixing ratio of dust is more stable with altitude. For fixed absolute differences between the SRIM and HYGRO cases, this sharp increase would cause a decrease in the relative differences in mass mixing ratios towards the surface. In the case of BC and POA, the

absolute differences between simulations also decrease towards the surface. In the case of dust, the absolute differences between simulations appear to decrease with altitude. The available profiles therefore suggest that the relative differences in surface concentrations between simulations are smaller for BC, POA, and SOA than the relative differences in the total columns would be, while the opposite may be the case for dust. Given that we do normalise by the surface concentrations, this means that the relative differences in normalised wet deposition may be overstated for BC, POA, and SOA, and understated for dust relative to wet deposition normalised by the total burden.

[Figure]

[Figure]

[Figure]

[Figure]

**Line 269, "We convert observed organic carbon to organic matter assuming a mass-to-carbon ratio of 1.8". Could the authors show references (ratio of 1.8) here?**

We have revised this sentence to the following to include the reference for our assumption:

*We convert observed organic carbon to organic matter assuming a mass-to-carbon ratio of 1.8, consistent with the assumption routinely used to calculate aerosol extinction based on the IMPROVE data (Pitchford et al., 2007).*

***Lines 315-322 and Figure 4, it is difficult to understand the BC transport and evaporation from Figure 4. Could the authors explain more clearly? How about the changes in BC column over the source (forest fires) and downwind regions?***

Unfortunately, due to the large size of our output files, we did not save model output of the total aerosol column. However, to help explain our point, we add the following figure to the supplement as Fig. S7. The figure is as Fig. 4, but for the highest model level for which we saved data (about 185 hPa above the surface).

[Figure]

We also add the following text to Sect. 3.1.4:

*If we examine the differences in BC concentrations at a higher altitude typically above clouds, as shown at about 185 hPa above the surface in Fig. S7, we do not see these decreases, but we do see that BC from these wildfires has been lofted to this altitude. [...] For this reason, the peaks in absolute differences are further downwind at the surface, in Ontario and Quebec (Fig. 4), than at about 185 hPa above the surface, where the peaks in absolute differences are in Nunavut and above Hudson's Bay (Fig. S7).*

***Lines 374-379, why AAOD decrease in the Gulf of Mexico and just south the Great Lakes region? Are these regions close to the anthropogenic source regions? Is the resolving the coating thickness of BC important for freshly emitted anthropogenic BC particles? Could you add explanations?***

First, we note that we had written the Gulf of Mexico in error. We meant the Gulf of California. We apologise for the mistake. We have added the following to the text to explain why the decreases in AAOD due to coating thickness outweigh the increases in AAOD due to less wet deposition in this region:

*As seen in Fig. 2, wet deposition was relatively low during our simulations in both of these regions, especially around the Gulf of California. This is due largely to less cloudiness and to less precipitation, which is shown in Fig. 10. Also, both of these regions have greater emissions of anthropogenic BC, as can be seen in Fig. S9.*

***It is helpful to show the emission map of BC (both anthropogenic and forest fires) during the model simulation period in Supplemental figures to clarify the source and remote regions.***

All emissions in GEM-MACH are classified as either area emissions or major point-source emissions, with wildfire emissions being handled separately from each of these through the CFFEPS system. We show below the mean BC emissions from each of these three sources during July 2016. The wildfire and major point-source emissions are displayed on a 0.25° latitude by 0.25° longitude grid to allow the colours of isolated point sources to be visible. Note that the same source sector (e.g. ocean shipping) can be classified as a major point source in some regions and an area source in others. We have added this figure to the supplement.

[Figure]

***Technical comments:***
***Table 3, I could not find the word "Table 3" in the text.***

We thank the referee for noting that we did not refer to this table, we now do so in Sect. 3.1.2.

***Lines 92-93, "0-2.5 um and 2.5-10 um". Are these diameters?***

Thank you, yes, these are aerosol dry diameters. We now clarify this.

***Line 101, "in that we our BC-free", remove "we", typo?***

Thank you, we have corrected this.

***Line 310, "BC concentrations at the surface". BC mixing ratio?***

Thank you, we have corrected this here, in the figure caption, and elsewhere where it would be more precise to refer to the BC mixing ratio.

***Figure S1, please add units, like Figure 1.***

Thank you, we have added the units.

References:

[revised manuscript text omitted]